# Learning Graphon Mean Field Games and Approximate Nash Equilibria

**Kai Cui & Heinz Koeppl**
Department of Electrical Engineering, Technische Universität Darmstadt, Germany
`{kai.cui,heinz.koeppl}@bcs.tu-darmstadt.de`

## Abstract

Recent advances at the intersection of dense large graph limits and mean field games have begun to enable the scalable analysis of a broad class of dynamical sequential games with large numbers of agents. So far, results have been largely limited to graphon mean field systems with continuous-time diffusive or jump dynamics, typically without control and with little focus on computational methods. We propose a novel discrete-time formulation for graphon mean field games as the limit of non-linear dense graph Markov games with weak interaction. On the theoretical side, we give extensive and rigorous existence and approximation properties of the graphon mean field solution in sufficiently large systems. On the practical side, we provide general learning schemes for graphon mean field equilibria by either introducing agent equivalence classes or reformulating the graphon mean field system as a classical mean field system. By repeatedly finding a regularized optimal control solution and its generated mean field, we successfully obtain plausible approximate Nash equilibria in otherwise infeasible large dense graph games with many agents. Empirically, we are able to demonstrate on a number of examples that the finite-agent behavior comes increasingly close to the mean field behavior for our computed equilibria as the graph or system size grows, verifying our theory. More generally, we successfully apply policy gradient reinforcement learning in conjunction with sequential Monte Carlo methods.

## 1 Introduction

Today, reinforcement learning (RL) finds application in various application areas such as robotics (Kober et al., 2013), autonomous driving (Kiran et al., 2021) or navigation of stratospheric balloons (Bellemare et al., 2020) as a method to realize effective sequential decision-making in complex problems. RL remains a very active research area, and there remain many challenges in multi-agent reinforcement learning (MARL) as a generalization of RL such as learning goals, non-stationarity and scalability of algorithms (Zhang et al., 2021). Nonetheless, potential applications for MARL are manifold and include e.g. teams of unmanned aerial vehicles (Tožička et al., 2018; Pham et al., 2018) or video games (Berner et al., 2019; Vinyals et al., 2017). While the domain of MARL is somewhat empirically successful, problems quickly become intractable as the number of agents grows, and methods in MARL typically miss a theoretical foundation. A recent tractable approach to handling the scalability problem in MARL are competitive mean field games and cooperative mean field control (Gu et al., 2021). Instead of considering generic multi agent Markov games, one considers many agents under the weak interaction principle, i.e. each agent alone has a negligible influence on all other agents. This class of models naturally contains a large number of real world scenarios and can find application e.g. in analysis of power network resilience (Bagagiolo & Bauso, 2014), smart heating (Kizilkale & Malhame, 2014), edge computing (Banez et al., 2019) or flocking (Perrin et al., 2021b). See also Djehiche et al. (2017) for a review of other engineering applications.

**Mean field games.** Mean field games (MFGs) were first popularized in the independent seminal works of Huang et al. (2006) and Lasry & Lions (2007) for the setting of differential games with diffusion-type dynamics given by stochastic differential equations. See also Guéant et al. (2011); Bensoussan et al. (2013) for a review. Since then, extensions have been manifold and include e.g. discrete-time (Saldi et al., 2018), partial observability (Saldi et al., 2019), major-minor formulations

(Nourian & Caines, 2013) and many more. In the learning community, there has been immense recent interest in finding and analyzing solution methods for mean field equilibria (Cardaliaguet & Hadikhanloo, 2017; Mguni et al., 2018; Guo et al., 2019; Subramanian & Mahajan, 2019; Elie et al., 2020; Cui & Koeppl, 2021; Pasztor et al., 2021; Perolat et al., 2021; Perrin et al., 2021a), solving the inverse reinforcement learning problem (Yang et al., 2018a) or applying related approximations directly to MARL (Yang et al., 2018b). In contrast to our work, Yang et al. (2018a) consider states instead of agents on a graph, while Yang et al. (2018b) requires restrictive assumptions and only considers averages instead of distributions of the neighbors. Recently, focus increased also on the cooperative case of mean field control (Carmona et al., 2019b; Mondal et al., 2021), for which dynamic programming holds on an enlarged state space, resulting in a high-dimensional Markov decision process (Pham & Wei, 2018; Motte & Pham, 2019; Gu et al., 2020; Cui et al., 2021).

**Mean field systems on graphs.** For mean field systems on dense graphs, prior work mostly considers mean field systems without control (Vizuete et al., 2020) or time-dynamics, i.e. the static case (Parise & Ozdaglar, 2019; Carmona et al., 2019a). To the best of our knowledge, Gao & Caines (2017) and Caines & Huang (2019) are the first to consider general continuous-time diffusion-type graphon mean field systems with control, the latter proposing many clusters of agents as well as proving an approximate Nash property as the number of clusters and agents grows. There have since been efforts to control cooperative graphon mean field systems with diffusive linear dynamics using spectral methods (Gao & Caines, 2019a;b). On the other hand, Bayraktar et al. (2020); Bet et al. (2020) consider large non-clustered systems in a continuous-time diffusion-type setting without control, while Aurell et al. (2021b) and Aurell et al. (2021a) consider continuous-time linear-quadratic systems and continuous-time jump processes respectively. To the best of our knowledge, only Vasal et al. (2021) have considered solving and formulating a graphon mean field game in discrete time, though requiring analytic computation of an infinite-dimensional value function defined over all mean fields and thus being inapplicable to arbitrary problems in a black-box, learning manner. In contrast, we give a general learning scheme and also provide extensive theoretical analysis of our algorithms and (slightly different) model. Finally, for sparse graphs there exist preliminary results (Gkogkas & Kuehn, 2020; Lacker & Soret, 2020), though the setting remains to be developed.

**Our contribution.** In this work, we propose a dense graph limit extension of MFGs in discrete time, combining graphon mean field systems with mean field games. More specifically, we consider limits of many-agent systems with discrete-time graph-based dynamics and weak neighbor interactions. In contrast to prior works, we consider one of the first general discrete-time formulations as well as its controlled case, which is a natural setting for many problems that are inherently discrete in time or to be controlled digitally at discrete decision times. Our contribution can be summarized as: (i) formulating one of the first general discrete-time graphon MFG frameworks for approximating otherwise intractable large dense graph games; (ii) providing an extensive theoretical analysis of existence and approximation properties in such systems; (iii) providing general learning schemes for finding graphon mean field equilibria, and (iv) empirically evaluating our proposed approach with verification of theoretical results in the finite $N$-agent graph system, finding plausible approximate Nash equilibria for otherwise infeasible large dense graph games with many agents.

## 2 DENSE GRAPH MEAN FIELD GAMES

In the following, we will give a dense graph $N$-agent model as well as its corresponding mean field system, where agents are affected only by the overall state distribution of all neighbors, as visualized in Figure 1. As a result of the law of large numbers, this distribution will become deterministic – the mean field – as $N \to \infty$. We begin with graph-theoretical preliminaries, see also Lovász (2012) for a review. The study of dense large graph limits deals with the limiting representation of adjacency matrices called graphons. Define $\mathcal{I} := [0, 1]$ and $\mathcal{W}_0$ as the space of all bounded, symmetric and measurable functions (graphons) $W \in \mathcal{W}_0$, $W : \mathcal{I} \times \mathcal{I} \to \mathbb{R}$ bounded by $0 \le W \le 1$. For any simple graph $G = (\{1, \dots, N\}, \mathcal{E})$, we define its step-graphon a.e. uniquely by

$$W_G(x, y) = \sum_{i,j \in \{1,\dots,N\}} \mathbf{1}_{(i,j) \in \mathcal{E}} \cdot \mathbf{1}_{x \in (\frac{i-1}{N}, \frac{i}{N}]} \cdot \mathbf{1}_{y \in (\frac{j-1}{N}, \frac{j}{N}]}, \tag{1}$$

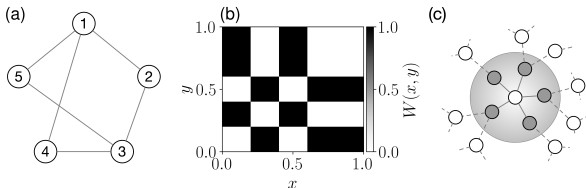

Figure 1: Graphical model visualization. **(a)**: A graph with 5 nodes; **(b)**: The associated step graphon of the graph in (a) as a continuous domain version of its adjacency matrix; **(c)**: A visualization of the dynamics, i.e. the center agent is affected only by its neighbors (grey).

see e.g. Figure 1. We equip $\mathcal{W}_0$ with the cut (semi-)norm $\|\cdot\|_\square$ and cut (pseudo-)metric $\delta_\square$

$$\|W\|_\square := \sup_{S,T} \left| \int_{S \times T} W(x,y)\, \mathrm{d}x\, \mathrm{d}y \right|, \quad \delta_\square(W,W') := \inf_\varphi \|W - W'_\varphi\|_\square, \tag{2}$$

for graphons $W, W' \in \mathcal{W}_0$ and $W'_\varphi(x,y) := W'(\varphi(x), \varphi(y))$, where the supremum is over all measurable subsets $S, T \subseteq \mathcal{I}$ and the infimum is over measure-preserving bijections $\varphi \colon \mathcal{I} \to \mathcal{I}$.

To provide motivation, note that convergence in $\delta_\square$ is equivalent to e.g. convergence of probabilities of locally encountering any fixed subgraph by randomly sampling a subset of nodes. Many such properties of graph sequences $(G_N)_{N \in \mathbb{N}}$ converging to some graphon $W \in \mathcal{W}_0$ can then be described by $W$, and we point to Lovász (2012) for details. In this work, we will primarily use the analytical fact that for converging graphon sequences $\|W_{G_N} - W\|_\square \to 0$, we equivalently have

$$\|W_{G_N} - W\|_{L_\infty \to L_1} = \sup_{\|g\|_\infty \leq 1} \int_\mathcal{I} \left| \int_\mathcal{I} (W_{G_N}(\alpha, \beta) - W(\alpha, \beta)) g(\beta)\, \mathrm{d}\beta \right| \mathrm{d}\alpha \to 0 \tag{3}$$

under the operator norm of operators $L_\infty \to L_1$, see e.g. Lovász (2012), Lemma 8.11.

By Lovász (2012), Theorem 11.59, the above is equivalent to convergence in the cut metric $\delta_\square(W_{G_N}, W) \to 0$ up to relabeling. In the following, we will therefore assume sequences of simple graphs $G_N = (\mathcal{V}_N, \mathcal{E}_N)$ with vertices $\mathcal{V}_N = \{1, \ldots, N\}$, edge sets $\mathcal{E}_N$, edge indicator variables $\xi_{i,j}^N := \mathbf{1}_{(i,j) \in \mathcal{E}_N}$ for all nodes $i, j \in \mathcal{V}_N$, and associated step graphons $W_N$ converging in cut norm.

**Assumption 1.** *The sequence of step-graphons $(W_N)_{N \in \mathbb{N}}$ converges in cut norm $\|\cdot\|_\square$ or equivalently in operator norm $\|\cdot\|_{L_\infty \to L_1}$ as $N \to \infty$ to some graphon $W \in \mathcal{W}_0$, i.e.*

$$\|W_N - W\|_\square \to 0, \quad \|W_N - W\|_{L_\infty \to L_1} \to 0. \tag{4}$$

Next, we define $W$-random graphs to consist of vertices $\mathcal{V}_N := \{1, \ldots, N\}$ with adjacency matrices $\boldsymbol{\xi}^N$ generated by sampling graphon indices $\alpha_i$ uniformly from $\mathcal{I}$ and edges $\xi_{i,j}^N \sim$ Bernoulli$(W(\alpha_i, \alpha_j))$ for all vertices $i, j \in \mathcal{V}_N$. For experiments, by Lovász (2012), Lemma 10.16, we can thereby generate a.s. converging graph sequences by sampling $W$-random graphs for any fixed graphon $W \in \mathcal{W}_0$. In principle, one could also consider arbitrary graph generating processes whenever a valid relabeling function $\varphi$ is known.

In our work, the usage of graphons enables us to find mean field systems on dense graphs and to extend the expressiveness of classical MFGs. As examples, we will use the limiting graphons of uniform attachment, ranked attachment and $p$-Erdős–Rényi (ER) random graphs given by $W_{\mathrm{unif}}(x,y) = 1 - \max(x,y)$, $W_{\mathrm{rank}}(x,y) = 1 - xy$ and $W_{\mathrm{er}}(x,y) = p$ respectively (Borgs et al., 2011; Lovász, 2012), each of which exhibits different node connectivities as shown in Figure 2.

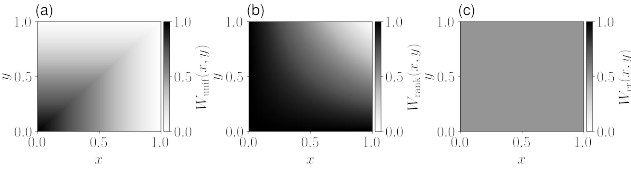

Figure 2: Three graphons used in our experiments. **(a)**: Uniform attachment graphon; **(b)**: Ranked attachment graphon; **(c)**: Erdős–Rényi (ER) graphon with edge probability 0.5.

**Finite agent graph game.** For simplicity of analysis, we consider finite state and action spaces $\mathcal{X}, \mathcal{U}$ as well as times $\mathcal{T} := \{0, 1, \ldots, T-1\}$. On a metric space $\mathcal{A}$, define the spaces of all Borel probability measures $\mathcal{P}(\mathcal{A})$ and all Borel measures $\mathcal{B}_1(\mathcal{A})$ bounded by 1, equipped with the $L_1$ norm. For simplified notation, we denote both a measure $\nu$ and its probability mass function by $\nu(\cdot)$. Define the space of policies $\Pi := \mathcal{P}(\mathcal{U})^{\mathcal{T} \times \mathcal{X}}$, i.e. agents apply Markovian feedback policies $\pi^i = (\pi_t^i)_{t \in \mathcal{T}} \in \Pi$ that act on local state information. This allows for the definition of weakly interacting agent state and action random variables

$$X_0^i \sim \mu_0, \quad U_t^i \sim \pi_t^i(\cdot \mid X_t^i), \quad X_{t+1}^i \sim P(\cdot \mid X_t^i, U_t^i, \mathbb{G}_t^i), \quad \forall t \in \mathcal{T}, \forall i \in \mathcal{V}_N \tag{5}$$

under some transition kernel $P: \mathcal{X} \times \mathcal{U} \times \mathcal{B}_1(\mathcal{X}) \to \mathcal{P}(\mathcal{X})$, where the empirical neighborhood mean field $\mathbb{G}_t^i$ of agent $i$ is defined as the $\mathcal{B}_1(\mathcal{X})$-valued (unnormalized) neighborhood state distribution

$$\mathbb{G}_t^i := \frac{1}{N} \sum_{j \in \mathcal{V}_N} \xi_{i,j}^N \delta_{X_t^j}, \tag{6}$$

where $\delta$ is the Dirac measure, i.e. each agent affects each other at most negligibly with factor $1/N$. Finally, for each agent $i$ we define separate, competitive objectives

$$J_i^N(\pi^1, \ldots, \pi^N) := \mathbb{E}\left[\sum_{t=0}^{T-1} r(X_t^i, U_t^i, \mathbb{G}_t^i)\right] \tag{7}$$

to be maximized over $\pi^i$, where $r: \mathcal{X} \times \mathcal{U} \times \mathcal{B}_1(\mathcal{X}) \to \mathbb{R}$ is an arbitrary reward function.

**Remark 1.** *We can also consider infinite horizons, $\tilde{J}_i^N(\pi^1, \ldots, \pi^N) \equiv \mathbb{E}\left[\sum_{t=0}^{\infty} \gamma^t r(X_t^i, U_t^i, \mathbb{G}_t^i)\right]$ with all results but Theorem 4 holding. One may also extend to state-action distributions, heterogeneous starting conditions and time-dependent $r$, $P$, though we avoid this for expositional simplicity.*

**Remark 2.** *Note that it is straightforward to extend to heterogeneous agents by modelling agent types as part of the agent state, see also e.g. Mondal et al. (2021). It is only required to model agent states in a unified manner, which does not imply that there can be no heterogeneity.*

With this, we can give a typical notion of Nash equilibria as found e.g. in Saldi et al. (2018). However, under graph convergence in Assumption 1, it is always possible for a finite number of nodes to have an arbitrary neighborhood differing from the graphon as $N \to \infty$. Thus, it is impossible to show approximate optimality for all nodes and only possible to show for an increasingly large fraction $1 - p \to 1$ of nodes. For this reason, we slightly weaken the notion of Nash equilibria by restricting to a fraction $1 - p$ of agents, as e.g. considered in (Carmona, 2004; Elie et al., 2020).

**Definition 1.** *An $(\varepsilon, p)$-Markov-Nash equilibrium (almost Markov-Nash equilibrium) for $\varepsilon, p > 0$ is defined as a tuple of policies $(\pi^1, \ldots, \pi^N) \in \Pi^N$ such that for any $i \in \mathcal{W}_N$, we have*

$$J_i^N(\pi^1, \ldots, \pi^N) \geq \sup_{\pi \in \Pi} J_i^N(\pi^1, \ldots, \pi^{i-1}, \pi, \pi^{i+1}, \ldots, \pi^N) - \varepsilon, \tag{8}$$

*for some set $\mathcal{W}_N \subseteq \mathcal{V}_N$ containing at least $\lfloor (1-p)N \rfloor$ agents, i.e. $|\mathcal{W}_N| \geq \lfloor (1-p)N \rfloor$.*

The minimal such $\varepsilon > 0$ for any fixed policy tuple (and typically $p = 0$) is also called its exploitability. Whilst we ordain $\varepsilon$-optimality only for a fraction $1 - p$ of agents, if the fraction $p$ is negligible, it will have negligible impact on other agents as a result of the weak interaction property. Thus, the solution will remain approximately optimal for almost all agents for sufficiently small $p$ regardless of the behavior of that fraction $p$ of agents. In the following, we will give a limiting system that shall provide $(\varepsilon, p)$-Markov-Nash equilibria with $\varepsilon, p \to 0$ as $N \to \infty$.

**Graphon mean field game.** The formal $N \to \infty$ limit of the $N$-agent game constitutes its graphon mean field game (GMFG), which shall be rigorously justified in Section 3. We define the space of measurable state marginal ensembles $\mathcal{M}_t := \mathcal{P}(\mathcal{X})^{\mathcal{I}}$ and measurable mean field ensembles $\boldsymbol{\mathcal{M}} := \mathcal{P}(\mathcal{X})^{\mathcal{T} \times \mathcal{I}}$, in the sense that $\alpha \mapsto \mu_t^\alpha(x)$ is measurable for any $\boldsymbol{\mu} \in \boldsymbol{\mathcal{M}}$, $t \in \mathcal{T}$, $x \in \mathcal{X}$. Similarly, we define the space of measurable policy ensembles $\boldsymbol{\Pi} \subseteq \Pi^{\mathcal{I}}$, i.e. with measurable $\alpha \mapsto \pi_t^\alpha(u \mid x)$ for any $\boldsymbol{\pi} \in \boldsymbol{\Pi}$, $t \in \mathcal{T}$, $x \in \mathcal{X}$, $u \in \mathcal{U}$.

In the GMFG, we will consider infinitely many agents $\alpha \in \mathcal{I}$ instead of the finitely many $i \in \mathcal{V}_N$. As a result, we will have infinitely many policies $\pi^\alpha \in \Pi$ – one for each agent $\alpha$ – through some measurable policy ensemble $\boldsymbol{\pi} \in \boldsymbol{\Pi}$. We again define state and action random variables

$$X_0^\alpha \sim \mu_0, \quad U_t^\alpha \sim \pi_t^\alpha(\cdot \mid X_t^\alpha), \quad X_{t+1}^\alpha \sim P(\cdot \mid X_t^\alpha, U_t^\alpha, \mathbb{G}_t^\alpha), \quad \forall (\alpha, t) \in \mathcal{I} \times \mathcal{T} \tag{9}$$

where we introduce the (now deterministic) $\mathcal{B}_1(\mathcal{X})$-valued neighborhood mean field of agents $\alpha$ as

$$\mathbb{G}_t^\alpha := \int_{\mathcal{I}} W(\alpha, \beta) \mu_t^\beta \, \mathrm{d}\beta \tag{10}$$

for some deterministic $\boldsymbol{\mu} \in \mathcal{M}$. Under fixed $\boldsymbol{\pi} \in \boldsymbol{\Pi}$, $\mu_t^\alpha$ should be understood as the law of $X_t^\alpha$, $\mu_t^\alpha \equiv \mathcal{L}(X_t^\alpha)$. Finally, define the maximization objective of agent $\alpha$ over $\pi^\alpha$ for fixed $\boldsymbol{\mu} \in \mathcal{M}$ as

$$J_\alpha^{\boldsymbol{\mu}}(\pi^\alpha) \equiv \mathbb{E}\left[ \sum_{t=0}^{T-1} r(X_t^\alpha, U_t^\alpha, \mathbb{G}_t^\alpha) \right]. \tag{11}$$

To formulate the limiting version of Nash equilibria, we define a map $\Psi \colon \boldsymbol{\Pi} \to \mathcal{M}$ mapping from a policy ensemble $\boldsymbol{\pi} \in \boldsymbol{\Pi}$ to the corresponding generated mean field ensemble $\boldsymbol{\mu} = \Psi(\boldsymbol{\pi}) \in \mathcal{M}$ by

$$\mu_0^\alpha \equiv \mu_0, \quad \mu_{t+1}^\alpha(x') \equiv \sum_{x \in \mathcal{X}} \mu_t^\alpha(x) \sum_{u \in \mathcal{U}} \pi_t^\alpha(u \mid x) P(x' \mid x, u, \mathbb{G}_t^\alpha), \quad \forall \alpha \in \mathcal{I} \tag{12}$$

where integrability in (10) holds by induction, and note how then $\mu_t^\alpha = \mathcal{L}(X_t^\alpha)$.

Similarly, let $\Phi \colon \mathcal{M} \to 2^{\boldsymbol{\Pi}}$ map from a mean field ensemble $\boldsymbol{\mu}$ to the set of optimal policy ensembles $\boldsymbol{\pi}$ characterized by $\pi^\alpha \in \arg\max_{\pi \in \boldsymbol{\Pi}} J_\alpha^{\boldsymbol{\mu}}(\pi)$ for all $\alpha \in \mathcal{I}$, which is particularly fulfilled if $\pi_t^\alpha(u \mid x) > 0 \implies u \in \arg\max_{u' \in \mathcal{U}} Q_\alpha^{\boldsymbol{\mu}}(t, x, u')$ for all $\alpha \in \mathcal{I}, t \in \mathcal{T}, x \in \mathcal{X}, u \in \mathcal{U}$, where $Q_\alpha^{\boldsymbol{\mu}}$ is the optimal action value function under fixed $\boldsymbol{\mu} \in \mathcal{M}$ following the Bellman equation

$$Q_\alpha^{\boldsymbol{\mu}}(t, x, u) = r(x, u, \mathbb{G}_t^\alpha) + \sum_{x' \in \mathcal{X}} P(x' \mid x, u, \mathbb{G}_t^\alpha) \arg\max_{u' \in \mathcal{U}} Q_\alpha^{\boldsymbol{\mu}}(t+1, x', u') \tag{13}$$

with $Q_\alpha^{\boldsymbol{\mu}}(T, x, u) \equiv 0$ and generally time-dependent, see Puterman (2014) for a review.

We can now define the GMFG version of Nash equilibria as policy ensembles $\boldsymbol{\pi}$ generating mean field ensembles $\boldsymbol{\mu}$ under which they are optimal, as $\mu_t^\alpha = \mathcal{L}(X_t^\alpha)$ if all agents $\alpha \in \mathcal{I}$ follow $\pi^\alpha$.

**Definition 2.** *A Graphon Mean Field Equilibrium (GMFE) is a pair $(\boldsymbol{\pi}, \boldsymbol{\mu}) \in \boldsymbol{\Pi} \times \mathcal{M}$ such that $\boldsymbol{\pi} \in \Phi(\boldsymbol{\mu})$ and $\boldsymbol{\mu} = \Psi(\boldsymbol{\pi})$.*

## 3 THEORETICAL ANALYSIS

To obtain meaningful optimality results beyond empirical mean field convergence, we will need a Lipschitz assumption as in the uncontrolled, continuous-time case (cf. Bayraktar et al. (2020), Condition 2.3) and typical in mean field theory (Huang et al., 2006).

**Assumption 2.** *Let $r$, $P$, $W$ be Lipschitz continuous with Lipschitz constants $L_r, L_P, L_W > 0$.*

Note that all proofs but Theorem 1 also hold for only block-wise Lipschitz continuous $W$, see Appendix A.1. Since $\mathcal{X} \times \mathcal{U} \times B_1(\mathcal{X})$ is compact, $r$ is bounded by the extreme value theorem.

**Proposition 1.** *Under Assumption 2, $r$ will be bounded by $|r| \leq M_r$ for some constant $M_r > 0$.*

We then obtain existence of a GMFE by reformulating the GMFG as a classical MFG and applying existing results from Saldi et al. (2018). More precisely, we consider the equivalent MFG with extended state space $\mathcal{X} \times \mathcal{I}$, action space $\mathcal{U}$, policy $\tilde{\pi} \in \mathcal{P}(\mathcal{U})^{\mathcal{T} \times \mathcal{X} \times \mathcal{I}}$, mean field $\tilde{\mu} \in \mathcal{P}(\mathcal{X} \times \mathcal{I})^{\mathcal{T}}$, reward function $\tilde{r}((x, \alpha), u, \tilde{\mu}) := r(x, u, \int_{\mathcal{I}} W(\alpha_t, \beta) \tilde{\mu}_t(\cdot, \beta) \, \mathrm{d}\beta)$ and transition dynamics such that the states $(\tilde{X}_t, \alpha_t)$ follow $(\tilde{X}_0, \alpha_0) \sim \tilde{\mu}_0 := \mu_0 \otimes \mathrm{Unif}([0, 1])$ and

$$\tilde{U}_t \sim \tilde{\pi}_t(\cdot \mid \tilde{X}_t, \alpha_t), \quad \tilde{X}_{t+1} \sim P(\cdot \mid \tilde{X}_t, \tilde{U}_t, \int_{\mathcal{I}} W(\alpha_t, \beta) \tilde{\mu}_t(\cdot, \beta) \, \mathrm{d}\beta), \quad \alpha_{t+1} = \alpha_t. \tag{14}$$

**Theorem 1.** *Under Assumption 2, there exists a GMFE $(\boldsymbol{\pi}, \boldsymbol{\mu}) \in \boldsymbol{\Pi} \times \mathcal{M}$.*

Meanwhile, in finite games, even showing the existence of Nash equilibria in local feedback policies is problematic (Saldi et al., 2018). Note however, that while this reformulation will be useful for learning and existence, it does not allow us to conclude that the **finite graph** game is well approximated, as classical MFG approximation theorems e.g. in Saldi et al. (2018) do not consider the graph structure and directly use the limiting graphon $W$ in the dynamics (14).

As our next main result, we shall therefore show rigorously that the GMFE can provide increasingly good approximations of the $N$-agent finite graph game as $N \to \infty$. As mentioned, the following also holds for only block-wise Lipschitz continuous $W$ instead of fully Lipschitz continuous $W$. Complete mathematical proofs together with additional theoretical supplements can be found in Appendix A.1 and A.2. To obtain joint $N$-agent policies as approximate Nash equilibria from a GMFE $(\boldsymbol{\pi}, \boldsymbol{\mu})$, we define the map $\Gamma_N(\boldsymbol{\pi}) \coloneqq (\pi^1, \pi^2, \ldots, \pi^N) \in \Pi^N$, where

$$\pi_t^i(u \mid x) \coloneqq \pi_t^{\alpha_i}(u \mid x), \quad \forall(\alpha, t, x, u) \in \mathcal{I} \times \mathcal{T} \times \mathcal{X} \times \mathcal{U} \tag{15}$$

with $\alpha_i = \frac{i}{N}$, as by Assumption 1 the agents are correctly labeled such that they match up with their limiting graphon indices $\alpha_i \in \mathcal{I}$. In our experiments, we use the $\alpha_i$ generated during the generation process of the $W$-random graphs, though for arbitrary finite systems one would have to first identify the graphon as well as an appropriate assignment of agents to graphon indices $\alpha_i \in \mathcal{I}$, which is a separate, non-trivial problem requiring at least graphon estimation, e.g. Xu (2018).

For theoretical analysis, we propose to lift the empirical distributions and policy tuples to the continuous domain $\mathcal{I}$, i.e. under an $N$-agent policy tuple $(\pi^1, \ldots, \pi^N) \in \Pi^N$, we define the step policy ensemble $\boldsymbol{\pi}^N \in \boldsymbol{\Pi}$ and the random empirical step measure ensemble $\boldsymbol{\mu}^N \in \boldsymbol{\mathcal{M}}$ by

$$\pi_t^{N,\alpha} \coloneqq \sum_{i \in \mathcal{V}_N} \mathbf{1}_{\alpha \in (\frac{i-1}{N}, \frac{i}{N}]} \cdot \pi_t^i, \quad \mu_t^{N,\alpha} \coloneqq \sum_{i \in \mathcal{V}_N} \mathbf{1}_{\alpha \in (\frac{i-1}{N}, \frac{i}{N}]} \cdot \delta_{X_t^j}, \quad \forall(\alpha, t) \in \mathcal{I} \times \mathcal{T}. \tag{16}$$

In the following, we consider deviations of the $i$-th agent from $(\pi^1, \pi^2, \ldots, \pi^N) = \Gamma_N(\boldsymbol{\pi}) \in \Pi^N$ to $(\pi^1, \ldots, \pi^{i-1}, \hat{\pi}, \pi^{i+1}, \ldots, \pi^N) \in \Pi^N$, i.e. the $i$-th agent deviates by instead applying $\hat{\pi} \in \Pi$. Note that this includes the special case of no agent deviations. For any $f \colon \mathcal{X} \times \mathcal{I} \to \mathbb{R}$ and state marginal ensemble $\boldsymbol{\mu}_t \in \mathcal{M}_t$, define $\boldsymbol{\mu}_t(f) \coloneqq \int_{\mathcal{I}} \sum_{x \in \mathcal{X}} f(x, \alpha) \mu_t^\alpha(x) \, \mathrm{d}\alpha$. We are now ready to state our first result of convergence of empirical state distributions to the mean field, potentially at the classical rate $\mathcal{O}(1/\sqrt{N})$ and consistent with results in uncontrolled, continuous-time diffusive graphon mean field systems (cf. Bayraktar et al. (2020), Theorem 3.2).

**Theorem 2.** *Consider Lipschitz continuous $\boldsymbol{\pi} \in \boldsymbol{\Pi}$ up to a finite number of discontinuities $D_\pi$, with associated mean field ensemble $\boldsymbol{\mu} = \Psi(\boldsymbol{\pi})$. Under Assumption 1 and the $N$-agent policy $(\pi^1, \ldots, \pi^{i-1}, \hat{\pi}, \pi^{i+1}, \ldots, \pi^N) \in \Pi^N$ with $(\pi^1, \pi^2, \ldots, \pi^N) = \Gamma_N(\boldsymbol{\pi}) \in \Pi^N$, $\hat{\pi} \in \Pi$, $t \in \mathcal{T}$, we have for all measurable functions $f \colon \mathcal{X} \times \mathcal{I} \to \mathbb{R}$ uniformly bounded by some $M_f > 0$, that*

$$\mathbb{E}\left[\left|\boldsymbol{\mu}_t^N(f) - \boldsymbol{\mu}_t(f)\right|\right] \to 0 \tag{17}$$

*uniformly over all possible deviations $\hat{\pi} \in \Pi, i \in \mathcal{V}_N$. Furthermore, if the graphon convergence in Assumption 1 is at rate $\mathcal{O}(1/\sqrt{N})$, then this rate of convergence is also $\mathcal{O}(1/\sqrt{N})$.*

In particular, the technical Lipschitz requirement of $\boldsymbol{\pi}$ typically holds for neural-network-based policies (Mondal et al., 2021; Pasztor et al., 2021) and includes also the case of finitely many optimality regimes over all graphon indices $\alpha \in \mathcal{I}$, which is sufficient to achieve arbitrarily good approximate Nash equilibria through our algorithms as shown in Section 4. We would like to remark that the above result generalizes convergence of state histograms to the mean field solution, since the state marginals of agents are additionally close to each of their graphon mean field equivalents. The above will be necessary to show convergence of the dynamics of a deviating agent to

$$\hat{X}_0^{\frac{i}{N}} \sim \mu_0, \quad \hat{U}_t^{\frac{i}{N}} \sim \hat{\pi}_t(\cdot \mid \hat{X}_t^{\frac{i}{N}}), \quad \hat{X}_{t+1}^{\frac{i}{N}} \sim P(\cdot \mid \hat{X}_t^{\frac{i}{N}}, \hat{U}_t^{\frac{i}{N}}, \mathbb{G}^{\frac{i}{N}}), \quad \forall t \in \mathcal{T} \tag{18}$$

for almost all agents $i$, i.e. the dynamics are approximated by using the limiting deterministic neighborhood mean field $\mathbb{G}^{\frac{i}{N}}$, see Appendix A.1. This will imply the approximate Nash property:

**Theorem 3.** *Consider a GMFE $(\boldsymbol{\pi}, \boldsymbol{\mu})$ with Lipschitz continuous $\boldsymbol{\pi}$ up to a finite number of discontinuities $D_\pi$. Under Assumptions 1 and 2, for any $\varepsilon, p > 0$ there exists $N'$ such that for all $N > N'$, the policy $(\pi^1, \ldots, \pi^N) = \Gamma_N(\boldsymbol{\pi}) \in \Pi^N$ is an $(\varepsilon, p)$-Markov Nash equilibrium, i.e.*

$$J_i^N(\pi^1, \ldots, \pi^N) \geq \max_{\pi \in \Pi} J_i^N(\pi^1, \ldots, \pi^{i-1}, \pi, \pi^{i+1}, \ldots, \pi^N) - \varepsilon \tag{19}$$

*for all $i \in \mathcal{W}_N$ and some $\mathcal{W}_N \subseteq \mathcal{V}_N \colon |\mathcal{W}_N| \geq \lfloor (1-p)N \rfloor$.*

In general, Nash equilibria are highly intractable (Daskalakis et al., 2009). Therefore, solving the GMFG allows obtaining approximate Nash equilibria in the $N$-agent system for sufficiently large $N$, since $\varepsilon, p \to 0$ as $N \to \infty$. As a side result, we also obtain first results for the uncontrolled discrete-time case by considering trivial action spaces with $|\mathcal{U}| = 1$, see Corollary A.2 in the Appendix.

## 4 LEARNING GRAPHON MEAN FIELD EQUILIBRIA

By learning GMFE, one may potentially solve otherwise intractable large $N$-agent games. For learning, we can apply any existing techniques for classical MFGs (e.g. Mguni et al. (2018); Subramanian & Mahajan (2019); Guo et al. (2019)), since by (14) we have reformulated the GMFG as a classical MFG with extended state space. Nonetheless, it may make sense to treat the graphon index $\alpha \in \mathcal{I}$ separately, e.g. when treating special cases such as block graphons, or by grouping graphically similar agents. We repeatedly apply two functions $\hat{\Phi}, \hat{\Psi}$ by beginning with the mean field $\boldsymbol{\mu}^0 = \hat{\Psi}(\boldsymbol{\pi}^0)$ generated by the uniformly random policy $\boldsymbol{\pi}^0$, and computing $\boldsymbol{\pi}^{n+1} = \hat{\Phi}(\boldsymbol{\mu}^n)$, $\boldsymbol{\mu}^{n+1} = \hat{\Psi}(\boldsymbol{\pi}^{n+1})$ for $n = 0, 1, \ldots$ until convergence using one of the following two approaches:

- **Equivalence classes method.** We introduce agent equivalence classes, or discretization, of $\mathcal{I}$ for the otherwise uncountably many agents $\alpha \in \mathcal{I}$ by partitioning $\mathcal{I}$ into $M$ subsets. For example, in the special case of block graphons (block-wise constant $W$), one can solve separately for each block equivalence class (type) of agents, since all agents in the class share the same dynamics. Note that in contrast to multi-class MFGs (Huang et al., 2006), GMFGs are rigorously connected to finite graph games and can handle an uncountable number of classes $\alpha$. To deal with general graphons, we choose equidistant representatives $\alpha_i \in \mathcal{I}$, $i = 1, \ldots, M$ covering the whole interval $\mathcal{I}$, and approximate each agent $\alpha \in \tilde{\mathcal{I}}_i$ by the nearest $\alpha_i$ for the intervals $\tilde{\mathcal{I}}_i \subseteq \mathcal{I}$ of points closest to that $\alpha_i$ to obtain $M$ approximate equivalence classes. Formally, we approximate mean fields $\hat{\Psi}(\boldsymbol{\pi}) = \sum_{i=1}^M \mathbf{1}_{\alpha \in \tilde{\mathcal{I}}_i} \hat{\mu}^{\alpha_i}$ recursively computed over all times for any fixed policy ensemble $\boldsymbol{\pi}$, and similarly policies $\hat{\Phi}(\boldsymbol{\mu}) = \sum_{i=1}^M \mathbf{1}_{\alpha \in \tilde{\mathcal{I}}_i} \pi^{\alpha_i}$ where $\pi^{\alpha_i}$ is the optimal policy of $\alpha_i$ for fixed $\boldsymbol{\mu}$. We solve the optimal control problem for each equivalence class using backwards induction (alternatively, one may use reinforcement learning), and solve the evolution equation for the representatives $\alpha_i$ of the equivalence classes recursively. For space reasons, the details are found in Appendix A.3. Note that this does not mean that we consider the $N$-agent problem with $N = M$, but instead we approximate the limiting problem with the limiting graphon $W$, and the solution will be near-optimal for all sufficiently large finite systems at once.

- **Direct reinforcement learning.** We directly apply RL as $\hat{\Phi}$. The central idea is to consider the GMFG as a classical MFG with extended state space $\mathcal{X} \times \mathcal{I}$, i.e. for fixed mean fields, we solve the MDP defined by (14). Agents condition their policy not only on their own state, but also their node index $\alpha \in \mathcal{I}$ and the current time $t \in \mathcal{T}$, since the mean fields are non-stationary in general and require time-dependent policies for optimality. Here, we assume that we can sample from a simulator of (9) for a given fixed mean field as commonly assumed in MFG learning literature (Guo et al., 2019; Subramanian & Mahajan, 2019). For application to arbitrary finite systems, one could apply a model-based RL approach coupled with graphon estimation, though this remains outside the scope of this work. For solving the mean field evolution equation (12), we can again use any applicable numerical method and choose a conventional sequential Monte Carlo method for $\hat{\Psi}$. While it is possible to exactly solve optimal control problems for each agent equivalence class with finite state-action spaces, this is generally not the case for e.g. continuous state-action spaces. Here, a general reinforcement learning solution can solve otherwise intractable problems in an elegant manner, since the graphon index $\alpha$ simply becomes part of a continuous state space.

For convergence, we begin by stating the classical feedback regularity condition (Huang et al., 2006; Guo et al., 2019) after equipping $\boldsymbol{\Pi}, \mathcal{M}$ e.g. with the supremum metric.

**Proposition 2.** *Assume that the maps $\hat{\Psi}, \hat{\Phi}$ are Lipschitz with constants $c_1, c_2$ and $c_1 \cdot c_2 < 1$. Then the fixed point iteration $\boldsymbol{\mu}^{n+1} = \hat{\Psi}(\hat{\Phi}(\boldsymbol{\mu}^n))$ converges.*

Feedback regularity is not assured, and thus there is no general convergence guarantee. Whilst one could attempt to apply fictitious play (Mguni et al., 2018), additional assumptions will be needed for convergence. Instead, whenever necessary for convergence, we regularize by introducing Boltzmann policies $\pi_t^\alpha(u \mid x) \propto \exp(\frac{1}{\eta} Q_\alpha^{\boldsymbol{\mu}}(t, x, u))$ with temperature $\eta$, provably converging to an approximation for sufficiently high temperatures (Cui & Koeppl, 2021).

**Theorem 4.** *Under Assumptions 1 and 2, the equivalence classes algorithm with Boltzmann policies $\hat{\Phi}(\boldsymbol{\mu})_t^\alpha(u \mid x) \propto \exp(\frac{1}{\eta} Q_\alpha^{\boldsymbol{\mu}}(t, x, u))$ converges for sufficiently high temperatures $\eta > 0$.*

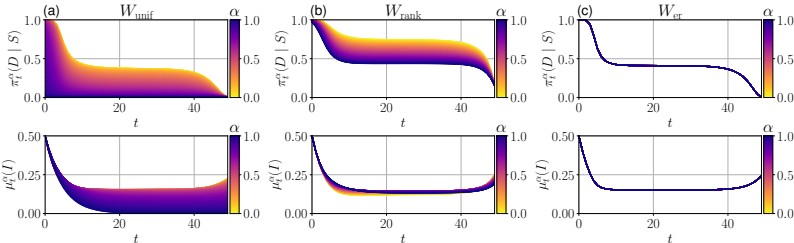

Figure 3: Achieved equilibrium via $M = 100$ approximate equivalence classes in SIS-Graphon, plotted for each agent $\alpha \in \mathcal{I}$. Top: Probability of taking precautions when healthy. Bottom: Probability of being infected. It can be observed that agents with less connections (higher $\alpha$) will take less precautions. **(a)**: Uniform attachment graphon; **(b)**: Ranked attachment graphon; **(c)**: ER graphon.

Importantly, even an exact solution of the GMFG only constitutes an approximate Nash equilibrium in the finite-graph system. Furthermore, even the existence of exact finite-system Nash equilibria in local feedback policies is not guaranteed, see the discussion in Saldi et al. (2018) and references therein. Therefore, little is lost by introducing slight additional approximations for the sake of a tractable solution, if at all needed (e.g. the Investment-Graphon problem in the following converges without introducing Boltzmann policies), since near optimality holds for small temperatures (Cui & Koeppl, 2021). Indeed, we find that we can show optimality of the equivalence classes approach for sufficiently fine partitions of $\mathcal{I}$, giving us a theoretical foundation for our proposed algorithms.

**Theorem 5.** *Under Assumptions 1 and 2, for a solution $(\boldsymbol{\pi}, \boldsymbol{\mu}) \in \boldsymbol{\Pi} \times \boldsymbol{\mathcal{M}}$, $\boldsymbol{\pi} \in \hat{\Phi}(\boldsymbol{\mu})$, $\boldsymbol{\mu} = \hat{\Psi}(\boldsymbol{\pi})$ of the $M$ equivalence classes method and for any $\varepsilon, p > 0$ there exists $N', M \in \mathbb{N}$ such that for all $N > N'$, the policy $(\pi^1, \ldots, \pi^N) = \Gamma_N(\boldsymbol{\pi}) \in \Pi^N$ is an $(\varepsilon, p)$-Markov Nash equilibrium.*

A theoretically rigorous analysis of the elegant direct reinforcement learning approach is beyond our scope and deferred to future works, though we empirically find that both methods agree.

## 5 EXPERIMENTS

In this section, we will give an empirical verification of our theoretical results. As we are unaware of any prior discrete-time GMFGs (except for the example in Vasal et al. (2021), which is similar to the first problem in the following), we propose two problems adapted from existing non-graph-based works on the three graphons in Figure 2. For space reasons, we defer detailed descriptions of problems and algorithms, plots as well as further analysis, including exploitability and a verification of stability of our solution with respect to the number of equivalence classes – to Appendix A.3.

The **SIS-Graphon** problem was considered in Cui & Koeppl (2021) as a classical discrete-time MFG. We impose an epidemics scenario where people (agents) are infected with probability proportional to the number of infected neighbors and recover with fixed probability. People may choose to take precautions (e.g. social distancing), avoiding potential costly infection periods at a fixed cost.

In the **Investment-Graphon** problem – an adaptation of a problem studied by Chen et al. (2021), where it was in turn adapted from Weintraub et al. (2010) – we consider many firms maximizing profits, where profits are proportional to product quality and decrease with total neighborhood product quality, i.e. the graph models overlap in e.g. product audience or functionality. Firms can invest to improve quality, though it becomes more unlikely to improve quality as their quality rises.

**Learned equilibrium behavior.** For the SIS-Graphon problem, we apply softmax policies for each approximate equivalence class to achieve convergence, see Appendix A.3 for details on temperature choice and influence. In Figure 3, the learned behavior can be observed for various $\alpha$. As expected, in the ER graphon case, behavior is identical over all $\alpha$. Otherwise, we find that agents take more precautions with many connections (low $\alpha$) than with few connections (high $\alpha$). For the uniform attachment graphon, we observe no precautions in case of negligible connectivity ($\alpha \to 1$), while for the ranked attachment graphon there is no such $\alpha \in \mathcal{I}$ (cf. Figure 2). Further, the fraction of infected agents at stationarity rises as $\alpha$ falls. A similar analysis holds for Investment-Graphon without need for regularization, see Appendix A.3.

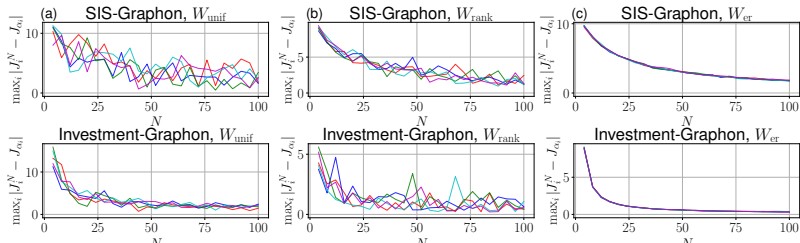

Figure 4: Decreasing maximum deviation between average $N$-agent objective and mean field objective over all agents for the GMFE policy and $5$ $W$-random graph sequences. **(a)**: Uniform attachment graphon; **(b)**: Ranked attachment graphon; **(c)**: ER graphon.

Note that the specific method of solution is not of central importance here, as in general any RL and filtering method can be substituted to handle 1. otherwise intractable or 2. inherently sample-based settings. Indeed, we achieve similar results using PPO (Schulman et al., 2017) in Investment-Graphon, enabling a general RL-based methodology for GMFGs. In Appendix A.3, we find that PPO achieves qualitatively and quantitatively similar behavior to the equivalence classes method, with slight deviations due to the approximations from PPO and Monte Carlo. In particular, the PPO exploitability $\varepsilon \approx 2$ remains low compared to $\varepsilon > 30$ for the uniform random policy, see Appendix A.3. In Appendix A.3, we also show how, due to the non-stationarity of the environment, a naive application of MARL (Yu et al., 2021) fails to converge, while existing mean field MARL techniques (Yang et al., 2018b) remain incomparable as agents must observe the average actions of all neighbors. On SIS-Graphon, we require softmax policies to achieve convergence, which is not possible with PPO as no $Q$-function is learned. In general, one could use entropy regularized policies, e.g. SAC (Haarnoja et al., 2018), or alternatively use any value-based reinforcement learning method, though an investigation of the best approach is outside of our scope.

**Quantitative verification of the mean field approximation.** To verify the rigorously established accuracy of our mean field system empirically, we will generate $W$-random graphs. Note that there are considerable difficulties associated with an empirical verification of (19), since 1. for any $N$ one must check the Nash property for (almost) all $N$ agents, 2. finding optimal $\hat{\pi}$ is intractable, as no dynamic programming principle holds on the non-Markovian local agent state, while acting on the full state fails by the curse of dimensionality, and 3. the inaccuracy from estimating all $J_i^N$, $i = 1, \ldots, N$ at once increases with $N$ due to variance, i.e. cost scales fast with $N$ for fixed variance. Instead, we verify (26) in Appendix A.1 using the GMFE policy on systems of up to $N = 100$ agents, i.e. $\hat{\pi} = \pi^{\alpha_i}$ for the closest $\alpha_i$ and comparing for all agents at once ($p = 0$). Shown in Figure 4, for $W$-random graph sequences, at each $N$ we performed $10000$ runs to estimate $\max_i |J_i^N - J_{\alpha_i}|$. We find that the maximum deviation between achieved returns and mean field return decreases as $N \to \infty$, verifying that we obtain an increasingly good approximation of the finite $N$-agent graph system. The oscillations in Figure 4 stem from the randomly sampled graphs.

## 6 CONCLUSION

In this work, we have formulated a new framework for dense graph-based dynamical games with the weak interaction property. On the theoretical side, we have given one of the first general discrete-time GMFG formulations with existence conditions and approximate Nash property of the finite graph system, thus extending classical MFGs and allowing for a tractable, theoretically well-founded solution of competitive large-scale graph-based games on large dense graphs. On the practical side, we have proposed a number of computational methods to tractably compute GMFE and experimentally verified the plausibility of our methodology on a number of examples. Venues for further extensions are manifold and include extensions of theory to e.g. continuous spaces, partial observability or common noise. So far, graphons assume dense graphs and cannot properly describe sparse graphs ($W = 0$), which remain an active frontier of research. Finally, real-world application scenarios may be of interest, where estimation of agent graphon indices becomes important for model-based MARL. We hope that our work inspires further applications and research into scalable MARL using graphical dynamical systems based on graph limit theory and mean field theory.

ACKNOWLEDGMENTS

This work has been funded by the LOEWE initiative (Hesse, Germany) within the emergenCITY center. The authors also acknowledge support by the German Research Foundation (DFG) via the Collaborative Research Center (CRC) 1053 – MAKI.

ETHICS STATEMENT

Existing mean field methodologies, including ours, currently require manual modeling and have not been applied in a model-based reinforcement learning setting for given finite agent systems. As a result, we do not foresee any immediate ethical issues stemming from this work.

REPRODUCIBILITY STATEMENT

For reproducibility, in the supplement we provide all code required to reproduce all results in this work. This includes but is not limited to our models and problems, algorithms as well as all plotting scripts for all of the figures found in this work.

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

## A APPENDIX

### A.1 THEORETICAL DETAILS

In this section, we will give all intermediate results required to prove the results in the main text, as well as additional results, e.g. for the uncontrolled case. For convenience, we first state all obtained theoretical result. Proofs for each of the theorems and corollaries can be found in their own sections further below.

Note that except for Theorem 1, as mentioned in the main text we can also slightly weaken Assumption 2 to block-wise Lipschitz continuous $W$, i.e. there exist $L_W > 0$ and disjoint intervals $\{\mathcal{I}_1, \ldots, \mathcal{I}_Q\}$, $\cup_i \mathcal{I}_i = \mathcal{I}$ s.t. $\forall i, j \in \{1, \ldots, Q\}$,

$$|W(x, y) - W(\tilde{x}, \tilde{y})| \le L_W(|x - \tilde{x}| + |y - \tilde{y}|), \quad \forall (x, y), (\tilde{x}, \tilde{y}) \in \mathcal{I}_i \times \mathcal{I}_j \tag{20}$$

which is fulfilled e.g. for block-wise Lipschitz-continuous or block-wise constant graphons.

For $\alpha \in \mathcal{I}$, define the $\alpha$-neighborhood maps $\mathbb{G}^\alpha \colon \mathcal{M}_t \to \mathcal{P}(\mathcal{X})$ and empirical $\alpha$-neighborhood maps $\mathbb{G}_N^\alpha \colon \mathcal{M}_t \to \mathcal{P}(\mathcal{X})$ as

$$\mathbb{G}^\alpha(\boldsymbol{\mu}_t) \coloneqq \int_{\mathcal{I}} W(\alpha, \beta) \mu_t^\beta \, \mathrm{d}\beta, \quad \mathbb{G}_N^\alpha(\boldsymbol{\mu}_t) \coloneqq \int_{\mathcal{I}} W_N(\alpha, \beta) \mu_t^\beta \, \mathrm{d}\beta \tag{21}$$

and note how we naturally have $\mathbb{G}_t^\alpha = \mathbb{G}^\alpha(\boldsymbol{\mu}_t)$ in the mean field system and $\mathbb{G}_t^i = \mathbb{G}_N^{\frac{i}{N}}(\boldsymbol{\mu}_t^N)$ in the finite system. Finally, for $\boldsymbol{\nu}, \boldsymbol{\nu}' \in \mathcal{M}_t$, $\boldsymbol{\pi} \in \Pi$ and graphon $W$, define the ensemble transition kernel operator $P_{t,\boldsymbol{\nu},W}^{\boldsymbol{\pi}} \colon \mathcal{M}_t \to \mathcal{M}_t$ via

$$\left(\boldsymbol{\nu} P_{t,\boldsymbol{\nu}',W}^{\boldsymbol{\pi}}\right)^\alpha \equiv \sum_{x \in \mathcal{X}} \nu^\alpha(x) \sum_{u \in \mathcal{U}} \pi_t^\alpha(u \mid x) P\left(\cdot \mid x, u, \int_{\mathcal{I}} W(\alpha, \beta) \nu'^\beta \, \mathrm{d}\beta\right) \tag{22}$$

and note how we have $\boldsymbol{\mu}_{t+1} = \boldsymbol{\mu}_t P_{t,\boldsymbol{\mu}_t,W}^{\boldsymbol{\pi}}$ in the mean field system.

After showing Theorem 2, we continue by showing convergence of the law of deviating agent state $X_t^i$ to the law of the corresponding auxiliary mean field systems given by

$$\hat{X}_0^{\frac{i}{N}} \sim \mu_0, \quad \hat{U}_t^{\frac{i}{N}} \sim \hat{\pi}_t(\cdot \mid \hat{X}_t^{\frac{i}{N}}), \quad \hat{X}_{t+1}^{\frac{i}{N}} \sim P(\cdot \mid \hat{X}_t^{\frac{i}{N}}, \hat{U}_t^{\frac{i}{N}}, \mathbb{G}_t^{\frac{i}{N}}), \quad \forall t \in \mathcal{T} \tag{23}$$

for almost all agents $i$ as $N \to \infty$.

**Lemma A.1.** *Consider Lipschitz continuous $\boldsymbol{\pi} \in \Pi$ up to a finite number of discontinuities $D_\pi$, with associated mean field ensemble $\boldsymbol{\mu} = \Psi(\boldsymbol{\pi})$. Under Assumptions 1 and 2 and the $N$-agent policy $(\pi^1, \ldots, \pi^{i-1}, \hat{\pi}, \pi^{i+1}, \ldots, \pi^N) \in \Pi^N$ where $(\pi^1, \pi^2, \ldots, \pi^N) = \Gamma_N(\boldsymbol{\pi}) \in \Pi^N$, $\hat{\pi} \in \Pi$ arbitrary, for any uniformly bounded family of functions $\mathcal{G}$ from $\mathcal{X}$ to $\mathbb{R}$ and any $\varepsilon, p > 0$, $t \in \mathcal{T}$, there exists $N' \in \mathbb{N}$ such that for all $N > N'$ we have*

$$\sup_{g \in \mathcal{G}} \left| \mathbb{E}\left[g(X_t^i)\right] - \mathbb{E}\left[g(\hat{X}_t^{\frac{i}{N}})\right] \right| < \varepsilon \tag{24}$$

*uniformly over $\hat{\pi} \in \Pi, i \in \mathcal{W}_N$ for some $\mathcal{W}_N \subseteq \mathcal{V}_N$ with $|\mathcal{W}_N| \ge \lfloor(1-p)N\rfloor$.*

*Similarly, for any uniformly Lipschitz, uniformly bounded family of measurable functions $\mathcal{H}$ from $\mathcal{X} \times \mathcal{B}_1(\mathcal{X})$ to $\mathbb{R}$ and any $\varepsilon, p > 0$, $t \in \mathcal{T}$, there exists $N' \in \mathbb{N}$ such that for all $N > N'$ we have*

$$\sup_{h \in \mathcal{H}} \left| \mathbb{E}\left[h(X_t^i, \mathbb{G}_N^{\frac{i}{N}}(\boldsymbol{\mu}_t^N))\right] - \mathbb{E}\left[h(\hat{X}_t^{\frac{i}{N}}, \mathbb{G}^{\frac{i}{N}}(\boldsymbol{\mu}_t))\right] \right| < \varepsilon \tag{25}$$

*uniformly over $\hat{\pi} \in \Pi, i \in \mathcal{W}_N$ for some $\mathcal{W}_N \subseteq \mathcal{V}_N$ with $|\mathcal{W}_N| \ge \lfloor(1-p)N\rfloor$.*

As a direct implication of the above results, the objective functions of almost all agents converge uniformly to the mean field objectives.

**Corollary A.1.** *Consider Lipschitz continuous $\boldsymbol{\pi} \in \Pi$ up to a finite number of discontinuities $D_\pi$, with associated mean field ensemble $\boldsymbol{\mu} = \Psi(\boldsymbol{\pi})$. Under Assumptions 1 and 2 and the $N$-agent policy $(\pi^1, \ldots, \pi^{i-1}, \hat{\pi}, \pi^{i+1}, \ldots, \pi^N) \in \Pi^N$ where $(\pi^1, \pi^2, \ldots, \pi^N) = \Gamma_N(\boldsymbol{\pi}) \in \Pi^N$, $\hat{\pi} \in \Pi$ arbitrary, for any $\varepsilon, p > 0$, there exists $N' \in \mathbb{N}$ such that for all $N > N'$ we have*

$$\left| J_i^N(\pi^1, \ldots, \pi^{i-1}, \hat{\pi}, \pi^{i+1}, \ldots, \pi^N) - J_{\frac{i}{N}}^{\boldsymbol{\mu}}(\hat{\pi}) \right| < \varepsilon \tag{26}$$

*uniformly over $\hat{\pi} \in \Pi, i \in \mathcal{W}_N$ for some $\mathcal{W}_N \subseteq \mathcal{V}_N$ with $|\mathcal{W}_N| \ge \lfloor(1-p)N\rfloor$.*

The approximate Nash property (Theorem 3) of a GMFE $(\boldsymbol{\pi}, \boldsymbol{\mu})$ then follows immediately from the definition of a GMFE, since $\boldsymbol{\pi}$ is by definition optimal under $\boldsymbol{\mu}$.

As a corollary, we also obtain results for the uncontrolled case without actions, which is equivalent to the case where $|\mathcal{U}| = 1$, i.e. there being only one trivial policy that is always optimal.

**Corollary A.2.** *Under Assumption 1 and $|\mathcal{U}| = 1$, we have for all measurable functions $f \colon \mathcal{X} \times \mathcal{I} \to \mathbb{R}$ uniformly bounded by $|f| \leq M_f$ and all $t \in \mathcal{T}$ that*

$$\mathbb{E}\left[\left|\boldsymbol{\mu}_t^N(f) - \boldsymbol{\mu}_t(f)\right|\right] \to 0. \tag{27}$$

*Furthermore, if the convergence in Assumption 1 is at rate $\mathcal{O}(1/\sqrt{N})$, the rate of convergence is also at $\mathcal{O}(1/\sqrt{N})$.*

*If further Assumption 2 holds, then for any uniformly bounded family of functions $\mathcal{G}$ from $\mathcal{X}$ to $\mathbb{R}$ and any $\varepsilon, p > 0$, $t \in \mathcal{T}$, there exists $N' \in \mathbb{N}$ such that for all $N > N'$ we have*

$$\sup_{g \in \mathcal{G}} \left| \mathbb{E}\left[g(X_t^i)\right] - \mathbb{E}\left[g(\hat{X}_t^{\frac{i}{N}})\right]\right| < \varepsilon \tag{28}$$

*uniformly over $i \in \mathcal{W}_N$ for some $\mathcal{W}_N \subseteq \mathcal{V}_N$ with $|\mathcal{W}_N| \geq \lfloor (1-p)N \rfloor$, and similarly for any uniformly Lipschitz, uniformly bounded family of measurable functions $\mathcal{H}$ from $\mathcal{X} \times \mathcal{B}_1(\mathcal{X})$ to $\mathbb{R}$ and any $\varepsilon, p > 0$, $t \in \mathcal{T}$, there exists $N' \in \mathbb{N}$ such that for all $N > N'$ we have*

$$\sup_{h \in \mathcal{H}} \left| \mathbb{E}\left[h(X_t^i, \mathbb{G}_N^{\frac{i}{N}}(\boldsymbol{\mu}_t^N))\right] - \mathbb{E}\left[h(\hat{X}_t^{\frac{i}{N}}, \mathbb{G}^{\frac{i}{N}}(\boldsymbol{\mu}_t))\right]\right| < \varepsilon \tag{29}$$

*uniformly over $i \in \mathcal{W}_N$ for some $\mathcal{W}_N \subseteq \mathcal{V}_N$ with $|\mathcal{W}_N| \geq \lfloor (1-p)N \rfloor$.*

## A.2 PROOFS

In this section, we will give full proofs to the statements made in the main text and in Appendix A.1.

### A.2.1 PROOF OF THEOREM 1

*Proof.* First, we will verify Saldi et al. (2018), Assumption 1 for the MFG with dynamics given by (14). For this purpose, as in Saldi et al. (2018) let us metrize the product space with the sup-metric, and equip the space $\mathcal{P}(\mathcal{X} \times \mathcal{I})$ with the weak topology. Note that the results hold for both the finite and infinite horizon setting, see Saldi et al. (2018), Remark 6.

(a) The reward function $\tilde{r}((x, \alpha), u, \tilde{\mu}) \coloneqq r(x, u, \int_{\mathcal{I}} W(\alpha_t, \beta)\tilde{\mu}_t(\cdot, \beta)\,\mathrm{d}\beta)$ is continuous, since for $((x_n, \alpha_n), u_n, \tilde{\mu}_n) \to ((x, \alpha), u, \tilde{\mu})$ we have

$$\int_{\mathcal{I}} W(\alpha_n, \beta)\tilde{\mu}_n(\cdot, \beta)\,\mathrm{d}\beta \to \int_{\mathcal{I}} W(\alpha, \beta)\tilde{\mu}(\cdot, \beta)\,\mathrm{d}\beta$$

by Lipschitz continuity of $W$ and weak convergence of $\tilde{\mu}_n$, and therefore

$$r\left(x_n, u_n, \int_{\mathcal{I}} W(\alpha_n, \beta)\tilde{\mu}_n(\cdot, \beta)\,\mathrm{d}\beta\right) \to r\left(x, u, \int_{\mathcal{I}} W(\alpha, \beta)\tilde{\mu}(\cdot, \beta)\,\mathrm{d}\beta\right)$$

by Assumption 2.

(b) The action space is compact and the state space is locally compact.

(c) Consider the moment function $w(x, \alpha) \equiv 2$. In this case, we can choose $\zeta = 1$ (we use $\zeta$ instead of $\alpha$ in Saldi et al. (2018)).

(d) The stochastic kernel $\tilde{P}$ that fulfills (14) such that $(\tilde{X}_{t+1}, \alpha_{t+1}) \sim \tilde{P}(\cdot \mid (\tilde{X}_t, \alpha_t), \tilde{U}_t, \tilde{\mu}_t)$ is weakly continuous, since for $((x_n, \alpha_n), u_n, \tilde{\mu}_n) \to ((x, \alpha), u, \tilde{\mu})$ we again have

$$\int_{\mathcal{I}} W(\alpha_n, \beta)\tilde{\mu}_n(\cdot, \beta)\,\mathrm{d}\beta \to \int_{\mathcal{I}} W(\alpha, \beta)\tilde{\mu}(\cdot, \beta)\,\mathrm{d}\beta$$

and therefore for any continuous bounded $f \colon \mathcal{X} \times \mathcal{I} \to \mathbb{R}$,

$$
\int_{\mathcal{X} \times \mathcal{I}} f \, \mathrm{d}\tilde{P}(\cdot \mid (x_n, \alpha_n), u_n, \tilde{\mu}_n) = \int_{\mathcal{I}} \int_{\mathcal{X}} f \, \mathrm{d}P^{\alpha}(\cdot \mid (x_n, \alpha), u_n, \tilde{\mu}_n) \, \delta_{\alpha_n}(\mathrm{d}\alpha)
$$

$$
= \int_{\mathcal{X}} f \, \mathrm{d}P(\cdot \mid x_n, u_n, \int_{\mathcal{I}} W(\alpha_n, \beta)\tilde{\mu}_n(\cdot, \beta) \, \mathrm{d}\beta)
$$

$$
\to \int_{\mathcal{X}} f \, \mathrm{d}P(\cdot \mid x, u, \int_{\mathcal{I}} W(\alpha, \beta)\tilde{\mu}(\cdot, \beta) \, \mathrm{d}\beta)
$$

$$
= \int_{\mathcal{X} \times \mathcal{I}} f \, \mathrm{d}\tilde{P}(\cdot \mid (x, \alpha), u, \tilde{\mu})
$$

by disintegration of $\tilde{P}$ and (14).

(e) By boundedness of $r$, we trivially have $v(x) \equiv 1 \leq \infty$.

(f) By boundedness of $r$, we can trivially choose $\beta = 1$ (we flip the usage of $\beta$ and $\gamma$).

(g) As a result of the above choices, $\zeta \gamma \beta = \gamma < 1$ trivially.

By Saldi et al. (2018), Theorem 3.3 we have the existence of a mean field equilibrium $(\tilde{\pi}, \tilde{\mu})$ with some Markovian feedback policy $\tilde{\pi}$ acting on the state $(\tilde{X}_t, \alpha_t)$. By defining the mean field and policy ensembles $\boldsymbol{\pi}, \boldsymbol{\mu}$ via $\pi_t^{\alpha}(u \mid x) = \tilde{\pi}_t(u \mid x, \alpha)$, $\mu_t^{\alpha} = \tilde{\mu}_t(\cdot, \alpha)$, we obtain existence of the $\alpha$-a.e. optimal policy ensemble $\boldsymbol{\pi}$, since at any time $t \in \mathcal{T}$, the joint state-action distribution $\tilde{\mu}_t \otimes \tilde{\pi}_t$ puts mass 1 on optimal state-action pairs (see Saldi et al. (2018), Theorem 3.6), implying that for a.e. $\alpha$ the policy must be optimal, as otherwise there exists a non-null set $\hat{\mathcal{I}}_0 \subseteq \mathcal{I}$ such that for all $\alpha \in \hat{\mathcal{I}}_0$, there is some suboptimality $\varepsilon > 0$, which directly contradicts the prequel.

For the remaining suboptimal $\alpha \in \mathcal{I}_0$ in the null set $\mathcal{I}_0 \subseteq \mathcal{I}$, we redefine $\boldsymbol{\pi}$ optimally for those $\alpha$ (always possible in our case, see e.g. Puterman (2014)). This policy ensemble generates $\boldsymbol{\mu} = \Psi(\boldsymbol{\pi})$ $\alpha$-a.e. uniquely, and we need only consider its $\alpha$-a.e. unique equivalence class for optimality, implying $\boldsymbol{\pi} \in \Phi(\boldsymbol{\mu})$. Furthermore, $\boldsymbol{\mu}$ is always measurable by definition, whereas $\boldsymbol{\pi}$ is measurable because $\tilde{\pi}_t$ is by definition a Markov kernel, and thus $\tilde{\pi}_t(u \mid \cdot, \cdot) = \tilde{\pi}_t(\{u\} \mid \cdot, \cdot)$ for Borel set $\{u\}$ is a measurable function, which implies measurability of $\tilde{\pi}_t(u \mid x, \cdot)$ (see e.g. Yeh (2014), Appendix E). Therefore, we have proven existence of the GMFE $(\boldsymbol{\pi}, \boldsymbol{\mu})$. $\qquad \square$

### A.2.2 PROOF OF THEOREM 2

*Proof.* The proof is by induction as follows.

**Initial case.** For $t = 0$, we trivially have for all measurable functions $f \colon \mathcal{X} \times \mathcal{I} \to \mathbb{R}$ uniformly bounded by $|f| \leq M_f$ a law of large numbers result

$$
\mathbb{E}\left[ |\boldsymbol{\mu}_0^N(f) - \boldsymbol{\mu}_0(f)| \right]
$$

$$
= \mathbb{E}\left[ \left| \int_{\mathcal{I}} \sum_{x \in \mathcal{X}} \mu_0^{N, \alpha}(x) \, f(x, \alpha) - \sum_{x \in \mathcal{X}} \mu_0^{\alpha}(x) \, f(x, \alpha) \, \mathrm{d}\alpha \right| \right]
$$

$$
= \mathbb{E}\left[ \left| \frac{1}{N} \sum_{i \in \mathcal{V}_N} \left( \int_{(\frac{i-1}{N}, \frac{i}{N}]} f(X_0^i, \alpha) \, \mathrm{d}\alpha - \mathbb{E}\left[ \int_{(\frac{i-1}{N}, \frac{i}{N}]} f(X_0^i, \alpha) \, \mathrm{d}\alpha \right] \right) \right| \right]
$$

$$
\leq \left( \mathbb{E}\left[ \left( \frac{1}{N} \sum_{i \in \mathcal{V}_N} \left( \int_{(\frac{i-1}{N}, \frac{i}{N}]} f(X_0^i, \alpha) \, \mathrm{d}\alpha - \mathbb{E}\left[ \int_{(\frac{i-1}{N}, \frac{i}{N}]} f(X_0^i, \alpha) \, \mathrm{d}\alpha \right] \right) \right)^2 \right] \right)^{\frac{1}{2}}
$$

$$
= \left( \frac{1}{N^2} \sum_{i \in \mathcal{V}_N} \mathbb{E}\left[ \left( \int_{(\frac{i-1}{N}, \frac{i}{N}]} f(X_0^i, \alpha) \, \mathrm{d}\alpha - \mathbb{E}\left[ \int_{(\frac{i-1}{N}, \frac{i}{N}]} f(X_0^i, \alpha) \, \mathrm{d}\alpha \right] \right)^2 \right] \right)^{\frac{1}{2}} \leq \frac{2M_f}{\sqrt{N}}
$$

by definition of $\boldsymbol{\mu}_t^N$, independence of $\{X_0^i\}_{i \in \mathcal{V}_N}$ and $X_0^i \sim \mu_0 = \mu_0^{\alpha}$ for all $i \in \mathcal{V}_N, \alpha \in \mathcal{I}$, where the second equality follows from Fubini's theorem.

**Induction step.** Assume that the induction assumption holds at $t$. Then by definition of $\boldsymbol{\mu}_t^N$, for all bounded function $f : \mathcal{X} \times \mathcal{I} \to \mathbb{R}$ with $|f| \le M_f$,

$$
\begin{aligned}
\mathbb{E}\left[\left|\boldsymbol{\mu}_{t+1}^N(f) - \boldsymbol{\mu}_{t+1}(f)\right|\right] &\le \mathbb{E}\left[\left|\boldsymbol{\mu}_{t+1}^N(f) - \boldsymbol{\mu}_t^N P_{t,\boldsymbol{\mu}_t^N,W_N}^{\boldsymbol{\pi}^N}(f)\right|\right] \\
&\quad + \mathbb{E}\left[\left|\boldsymbol{\mu}_t^N P_{t,\boldsymbol{\mu}_t^N,W_N}^{\boldsymbol{\pi}^N}(f) - \boldsymbol{\mu}_t^N P_{t,\boldsymbol{\mu}_t^N,W}^{\boldsymbol{\pi}^N}(f)\right|\right] \\
&\quad + \mathbb{E}\left[\left|\boldsymbol{\mu}_t^N P_{t,\boldsymbol{\mu}_t^N,W}^{\boldsymbol{\pi}^N}(f) - \boldsymbol{\mu}_t^N P_{t,\boldsymbol{\mu}_t^N,W}^{\boldsymbol{\pi}}(f)\right|\right] \\
&\quad + \mathbb{E}\left[\left|\boldsymbol{\mu}_t^N P_{t,\boldsymbol{\mu}_t^N,W}^{\boldsymbol{\pi}}(f) - \boldsymbol{\mu}_t^N P_{t,\boldsymbol{\mu}_t,W}^{\boldsymbol{\pi}}(f)\right|\right] \\
&\quad + \mathbb{E}\left[\left|\boldsymbol{\mu}_t^N P_{t,\boldsymbol{\mu}_t,W}^{\boldsymbol{\pi}}(f) - \boldsymbol{\mu}_{t+1}(f)\right|\right] .
\end{aligned}
$$

**First term.** We have by definition of $\boldsymbol{\mu}_t^N$

$$
\begin{aligned}
&\mathbb{E}\left[\left|\boldsymbol{\mu}_{t+1}^N(f) - \boldsymbol{\mu}_t^N P_{t,\boldsymbol{\mu}_t^N,W_N}^{\boldsymbol{\pi}^N}(f)\right|\right] \\
&= \mathbb{E}\left[\left|\int_{\mathcal{I}} \sum_{x \in \mathcal{X}} \mu_{t+1}^{N,\alpha}(x)\, f(x,\alpha)\, \mathrm{d}\alpha \right.\right. \\
&\qquad \left.\left. - \int_{\mathcal{I}} \sum_{x \in \mathcal{X}} \mu_t^{N,\alpha}(x) \sum_{u \in \mathcal{U}} \pi_t^{N,\alpha}(u \mid x) \sum_{x' \in \mathcal{X}} P\left(x' \mid x, u, \int_{\mathcal{I}} W_N(\alpha,\beta) \mu_t^{N,\beta}\, \mathrm{d}\beta\right) f(x',\alpha)\, \mathrm{d}\alpha \right|\right] \\
&= \mathbb{E}\left[\left|\frac{1}{N} \sum_{i \in \mathcal{V}_N}\left(\int_{(\frac{i-1}{N},\frac{i}{N}]} f(X_{t+1}^i,\alpha)\, \mathrm{d}\alpha - \mathbb{E}\left[\int_{(\frac{i-1}{N},\frac{i}{N}]} f(X_{t+1}^i,\alpha)\, \mathrm{d}\alpha \,\Big|\, \mathbf{X}_t\right]\right)\right|\right] \\
&\le \left(\mathbb{E}\left[\left(\frac{1}{N} \sum_{i \in \mathcal{V}_N}\left(\int_{(\frac{i-1}{N},\frac{i}{N}]} f(X_{t+1}^i,\alpha)\, \mathrm{d}\alpha - \mathbb{E}\left[\int_{(\frac{i-1}{N},\frac{i}{N}]} f(X_{t+1}^i,\alpha)\, \mathrm{d}\alpha \,\Big|\, \mathbf{X}_t\right]\right)\right)^2\right]\right)^{\frac{1}{2}} \\
&= \left(\frac{1}{N^2} \sum_{i \in \mathcal{V}_N} \mathbb{E}\left[\left(\int_{(\frac{i-1}{N},\frac{i}{N}]} f(X_{t+1}^i,\alpha)\, \mathrm{d}\alpha - \mathbb{E}\left[\int_{(\frac{i-1}{N},\frac{i}{N}]} f(X_{t+1}^i,\alpha)\, \mathrm{d}\alpha \,\Big|\, \mathbf{X}_t\right]\right)^2\right]\right)^{\frac{1}{2}} \\
&\le \frac{2M_f}{\sqrt{N}}
\end{aligned}
$$

where the last equality follows from conditional independence of $\{X_{t+1}^i\}_{i \in \mathcal{V}_N}$ given $\mathbf{X}_t \equiv \{X_t^i\}_{i \in \mathcal{V}_N}$ and the law of total expectation.

**Second term.** We have

$$
\begin{aligned}
&\mathbb{E}\left[\left|\boldsymbol{\mu}_t^N P_{t,\boldsymbol{\mu}_t^N,W_N}^{\boldsymbol{\pi}^N}(f) - \boldsymbol{\mu}_t^N P_{t,\boldsymbol{\mu}_t^N,W}^{\boldsymbol{\pi}^N}(f)\right|\right] \\
&= \mathbb{E}\left[\left|\int_{\mathcal{I}} \sum_{x \in \mathcal{X}} \mu_t^{N,\alpha}(x) \sum_{u \in \mathcal{U}} \pi_t^{N,\alpha}(u \mid x) \sum_{x' \in \mathcal{X}} P\left(x' \mid x, u, \int_{\mathcal{I}} W_N(\alpha,\beta) \mu_t^{N,\beta}\, \mathrm{d}\beta\right) f(x',\alpha)\, \mathrm{d}\alpha \right.\right. \\
&\qquad \left.\left. - \int_{\mathcal{I}} \sum_{x \in \mathcal{X}} \mu_t^{N,\alpha}(x) \sum_{u \in \mathcal{U}} \pi_t^{N,\alpha}(u \mid x) \sum_{x' \in \mathcal{X}} P\left(x' \mid x, u, \int_{\mathcal{I}} W(\alpha,\beta) \mu_t^{N,\beta}\, \mathrm{d}\beta\right) f(x',\alpha)\, \mathrm{d}\alpha \right|\right] \\
&\le |\mathcal{X}| M_f L_P\, \mathbb{E}\left[\int_{\mathcal{I}} \left\|\int_{\mathcal{I}} W_N(\alpha,\beta) \mu_t^{N,\beta}\, \mathrm{d}\beta - \int_{\mathcal{I}} W(\alpha,\beta) \mu_t^{N,\beta}\, \mathrm{d}\beta\right\|\, \mathrm{d}\alpha\right] \\
&\le |\mathcal{X}|^2 M_f L_P \sup_{x \in \mathcal{X}} \mathbb{E}\left[\int_{\mathcal{I}} \left|\int_{\mathcal{I}} W_N(\alpha,\beta) \mu_t^{N,\beta}(x) - W(\alpha,\beta) \mu_t^{N,\beta}(x)\, \mathrm{d}\beta\right|\, \mathrm{d}\alpha\right] \to 0
\end{aligned}
$$

by Assumption 1 and $\mu_t^{N,\beta}(x)$ trivially being bounded by 1. If the convergence in Assumption 1 is at rate $\mathcal{O}(1/\sqrt{N})$, then this convergence is also at rate $\mathcal{O}(1/\sqrt{N})$.

**Third term.** We have

$$
\mathbb{E}\left[\left|\boldsymbol{\mu}_t^N P_{t,\boldsymbol{\mu}_t^N,W}^{\boldsymbol{\pi}^N}(f) - \boldsymbol{\mu}_t^N P_{t,\boldsymbol{\mu}_t^N,W}^{\boldsymbol{\pi}}(f)\right|\right]
$$

$$
= \mathbb{E}\left[\left|\int_{\mathcal{I}} \sum_{x\in\mathcal{X}} \mu_t^{N,\alpha}(x) \sum_{u\in\mathcal{U}} \pi_t^{N,\alpha}(u\mid x) \sum_{x'\in\mathcal{X}} P\left(x'\mid x,u,\int_{\mathcal{I}} W(\alpha,\beta)\mu_t^{N,\beta}\,\mathrm{d}\beta\right) f(x',\alpha)\,\mathrm{d}\alpha \right.\right.
$$

$$
\left.\left. - \int_{\mathcal{I}} \sum_{x\in\mathcal{X}} \mu_t^{N,\alpha}(x) \sum_{u\in\mathcal{U}} \pi_t^{\alpha}(u\mid x) \sum_{x'\in\mathcal{X}} P\left(x'\mid x,u,\int_{\mathcal{I}} W(\alpha,\beta)\mu_t^{N,\beta}\,\mathrm{d}\beta\right) f(x',\alpha)\,\mathrm{d}\alpha\right|\right]
$$

$$
\leq |\mathcal{X}||\mathcal{U}|M_f\,\mathbb{E}\left[\int_{\mathcal{I}} \left|\pi_t^{N,\alpha}(u\mid x) - \pi_t^{\alpha}(u\mid x)\right|\,\mathrm{d}\alpha\right]
$$

$$
= |\mathcal{X}||\mathcal{U}|M_f\,\mathbb{E}\left[\sum_{j\in\mathcal{V}_N\setminus\{i\}} \int_{(\frac{j-1}{N},\frac{j}{N}]} \left|\pi_t^{\frac{\lceil N\alpha\rceil}{N}}(u\mid x) - \pi_t^{\alpha}(u\mid x)\right|\,\mathrm{d}\alpha\right]
$$

$$
+ |\mathcal{X}||\mathcal{U}|M_f\,\mathbb{E}\left[\int_{(\frac{i-1}{N},\frac{i}{N}]} \left|\hat{\pi}_t(u\mid x) - \pi_t^{\alpha}(u\mid x)\right|\,\mathrm{d}\alpha\right]
$$

$$
\leq |\mathcal{X}||\mathcal{U}|M_f\cdot\frac{L_\pi}{N} + |\mathcal{X}||\mathcal{U}|M_f\cdot\frac{2|D_\pi|}{N} + |\mathcal{X}||\mathcal{U}|M_f\cdot\frac{2}{N}
$$

by assumption of Lipschitz continuous $\pi$ up to a finite number of discontinuities $D_\pi$ as well as the deviating agent $i$'s error term, for which the integrands are bounded by 2.

**Fourth term.** We have

$$
\mathbb{E}\left[\left|\boldsymbol{\mu}_t^N P_{t,\boldsymbol{\mu}_t^N,W}^{\boldsymbol{\pi}}(f) - \boldsymbol{\mu}_t^N P_{t,\boldsymbol{\mu}_t,W}^{\boldsymbol{\pi}}(f)\right|\right]
$$

$$
= \mathbb{E}\left[\left|\int_{\mathcal{I}} \sum_{x\in\mathcal{X}} \mu_t^{N,\alpha}(x) \sum_{u\in\mathcal{U}} \pi_t^{\alpha}(u\mid x) \sum_{x'\in\mathcal{X}} P\left(x'\mid x,u,\int_{\mathcal{I}} W(\alpha,\beta)\mu_t^{N,\beta}\,\mathrm{d}\beta\right) f(x',\alpha)\,\mathrm{d}\alpha\right.\right.
$$

$$
\left.\left. - \int_{\mathcal{I}} \sum_{x\in\mathcal{X}} \mu_t^{N,\alpha}(x) \sum_{u\in\mathcal{U}} \pi_t^{\alpha}(u\mid x) \sum_{x'\in\mathcal{X}} P\left(x'\mid x,u,\int_{\mathcal{I}} W(\alpha,\beta)\mu_t^{\beta}\,\mathrm{d}\beta\right) f(x',\alpha)\,\mathrm{d}\alpha\right|\right]
$$

$$
\leq M_f|\mathcal{X}|\,\mathbb{E}\left[\sup_{x,u} \int_{\mathcal{I}} \left|P\left(x'\mid x,u,\int_{\mathcal{I}} W(\alpha,\beta)\mu_t^{N,\beta}\,\mathrm{d}\beta\right)\right.\right.
$$

$$
\left.\left. - P\left(x'\mid x,u,\int_{\mathcal{I}} W(\alpha,\beta)\mu_t^{\beta}\,\mathrm{d}\beta\right)\right|\,\mathrm{d}\alpha\right]
$$

$$
\leq M_f|\mathcal{X}|L_P \sum_{x'\in\mathcal{X}} \mathbb{E}\left[\int_{\mathcal{I}} \left|\int_{\mathcal{I}} W(\alpha,\beta)\mu_t^{N,\beta}(x')\,\mathrm{d}\beta - \int_{\mathcal{I}} W(\alpha,\beta)\mu_t^{\beta}(x')\,\mathrm{d}\beta\right|\,\mathrm{d}\alpha\right]
$$

$$
\leq M_f|\mathcal{X}|^2 L_P\cdot\frac{C'(1)}{\sqrt{N}}
$$

in the case of rate $\mathcal{O}(1/\sqrt{N})$, or uniformly to zero otherwise, from Lipschitz $P$ by defining the functions $f'_{x',\alpha}(x,\beta) = W(\alpha,\beta)\cdot\mathbf{1}_{x=x'}$ for any $(x',\alpha)\in\mathcal{X}\times\mathcal{I}$ and using the induction assumption on $f'_{x',\alpha}$ to obtain

$$
\mathbb{E}\left[\int_{\mathcal{I}} \left|\int_{\mathcal{I}} W(\alpha,\beta)\mu_t^{N,\beta}(x')\,\mathrm{d}\beta - \int_{\mathcal{I}} W(\alpha,\beta)\mu_t^{\beta}(x')\,\mathrm{d}\beta\right|\,\mathrm{d}\alpha\right]
$$

$$
= \int_{\mathcal{I}} \mathbb{E}\left[\left|\int_{\mathcal{I}} W(\alpha,\beta)\mu_t^{N,\beta}(x')\,\mathrm{d}\beta - \int_{\mathcal{I}} W(\alpha,\beta)\mu_t^{\beta}(x')\,\mathrm{d}\beta\right|\right]\,\mathrm{d}\alpha
$$

$$
= \int_{\mathcal{I}} \mathbb{E}\left[\left|\boldsymbol{\mu}_t^N(f'_{x',\alpha}) - \boldsymbol{\mu}_t(f'_{x',\alpha})\right|\right]\,\mathrm{d}\alpha \leq \frac{C'(1)}{\sqrt{N}}
$$

for some $C'(1) > 0$ uniformly over all $f'$ bounded by 1 if the convergence in Assumption 1 is at rate $\mathcal{O}(1/\sqrt{N})$, or uniformly to zero otherwise.

**Fifth term.** We have

$$\mathbb{E}\left[\left|\boldsymbol{\mu}_t^N P_{t,\boldsymbol{\mu}_t,W}^{\boldsymbol{\pi}}(f) - \boldsymbol{\mu}_t P_{t,\boldsymbol{\mu}_t,W}^{\boldsymbol{\pi}}(f)\right|\right]$$

$$= \mathbb{E}\left[\left|\int_{\mathcal{I}} \sum_{x\in\mathcal{X}} \mu_t^{N,\alpha}(x) \sum_{u\in\mathcal{U}} \pi_t^\alpha(u\mid x) \sum_{x'\in\mathcal{X}} P\left(x'\mid x,u,\int_{\mathcal{I}} W(\alpha,\beta)\mu_t^\beta\,\mathrm{d}\beta\right) f(x',\alpha)\,\mathrm{d}\alpha\right.\right.$$

$$\left.\left. - \int_{\mathcal{I}} \sum_{x\in\mathcal{X}} \mu_t^\alpha(x) \sum_{u\in\mathcal{U}} \pi_t^\alpha(u\mid x) \sum_{x'\in\mathcal{X}} P\left(x'\mid x,u,\int_{\mathcal{I}} W(\alpha,\beta)\mu_t^\beta\,\mathrm{d}\beta\right) f(x',\alpha)\,\mathrm{d}\alpha\right|\right]$$

$$= \mathbb{E}\left[\left|\int_{\mathcal{I}} \sum_{x\in\mathcal{X}} \mu_t^{N,\alpha}(x) f'(x,\alpha)\,\mathrm{d}\alpha - \int_{\mathcal{I}} \sum_{x\in\mathcal{X}} \mu_t^\alpha(x) f'(x,\alpha)\,\mathrm{d}\alpha\right|\right]$$

$$= \mathbb{E}\left[\left|\boldsymbol{\mu}_t^N(f') - \boldsymbol{\mu}_t(f')\right|\right] \le \frac{C'(M_f)}{\sqrt{N}}\,.$$

in the case of rate $\mathcal{O}(1/\sqrt{N})$, or uniformly to zero otherwise, again by induction assumption applied to the function

$$f'(x,\alpha) = \sum_{u\in\mathcal{U}} \pi_t^\alpha(u\mid x) \sum_{x'\in\mathcal{X}} P\left(x'\mid x,u,\int_{\mathcal{I}} W(\alpha,\beta)\mu_t^\beta\,\mathrm{d}\beta\right) f(x',\alpha)$$

bounded by $M_f$. This completes the proof by induction. $\qquad\square$

### A.2.3 PROOF OF LEMMA A.1

*Proof.* First, we will show that (24) implies (25).

**Proof of** (24) $\implies$ (25). We consider a uniformly Lipschitz, uniformly bounded family of measurable functions $\mathcal{H}$ from $\mathcal{X} \times \mathcal{B}_1(\mathcal{X})$ to $\mathbb{R}$. Let $M_h$ be the uniform bound of functions in $\mathcal{H}$ and $L_h$ be the uniform Lipschitz constant. Then, for arbitrary $h \in \mathcal{H}$ we have

$$\left|\mathbb{E}\left[h(X_t^i, \mathbb{G}_N^{\frac{i}{N}}(\boldsymbol{\mu}_t^N))\right] - \mathbb{E}\left[h(\hat{X}_t^{\frac{i}{N}}, \mathbb{G}^{\frac{i}{N}}(\boldsymbol{\mu}_t))\right]\right|$$

$$= \left|\mathbb{E}\left[h(X_t^i, \mathbb{G}_N^{\frac{i}{N}}(\boldsymbol{\mu}_t^N))\right] - \mathbb{E}\left[h(X_t^i, \mathbb{G}_N^{\frac{i}{N}}(\boldsymbol{\mu}_t))\right]\right|$$

$$+ \left|\mathbb{E}\left[h(X_t^i, \mathbb{G}_N^{\frac{i}{N}}(\boldsymbol{\mu}_t))\right] - \mathbb{E}\left[h(X_t^i, \mathbb{G}^{\frac{i}{N}}(\boldsymbol{\mu}_t))\right]\right|$$

$$+ \left|\mathbb{E}\left[h(X_t^i, \mathbb{G}^{\frac{i}{N}}(\boldsymbol{\mu}_t))\right] - \mathbb{E}\left[h(\hat{X}_t^{\frac{i}{N}}, \mathbb{G}^{\frac{i}{N}}(\boldsymbol{\mu}_t))\right]\right|$$

which we will analyze in the following.

**First term.** We have

$$\left|\mathbb{E}\left[h(X_t^i, \mathbb{G}_N^{\frac{i}{N}}(\boldsymbol{\mu}_t^N))\right] - \mathbb{E}\left[h(X_t^i, \mathbb{G}_N^{\frac{i}{N}}(\boldsymbol{\mu}_t))\right]\right|$$

$$\le \mathbb{E}\left[\mathbb{E}\left[\left|h(X_t^i, \mathbb{G}_N^{\frac{i}{N}}(\boldsymbol{\mu}_t^N)) - h(X_t^i, \mathbb{G}_N^{\frac{i}{N}}(\boldsymbol{\mu}_t))\right|\,\Big|\,X_t^i\right]\right]$$

$$\le L_h \,\mathbb{E}\left[\left\|\mathbb{G}_N^{\frac{i}{N}}(\boldsymbol{\mu}_t^N) - \mathbb{G}_N^{\frac{i}{N}}(\boldsymbol{\mu}_t)\right\|\right]$$

$$= L_h \sum_{x\in\mathcal{X}} \mathbb{E}\left[\left|\int_{\mathcal{I}} W_N(\tfrac{i}{N},\beta)\mu_t^{N,\beta}(x)\,\mathrm{d}\beta - \int_{\mathcal{I}} W_N(\tfrac{i}{N},\beta)\mu_t^\beta(x)\,\mathrm{d}\beta\right|\right] \le \frac{C(1)}{\sqrt{N}}$$

by Theorem 2 applied to the functions $f'_{N,i,x}(x',\beta) = W_N(\tfrac{i}{N},\beta)\cdot\mathbf{1}_{x=x'}$ uniformly bounded by 1.

**Second term.** Similarly, we have

$$\left|\mathbb{E}\left[h(X_t^i, \mathbb{G}_N^{\frac{i}{N}}(\boldsymbol{\mu}_t))\right] - \mathbb{E}\left[h(X_t^i, \mathbb{G}^{\frac{i}{N}}(\boldsymbol{\mu}_t))\right]\right|$$

$$\le L_h \|\mathbb{G}_N^{\frac{i}{N}}(\boldsymbol{\mu}_t) - \mathbb{G}^{\frac{i}{N}}(\boldsymbol{\mu}_t)\|_1$$

$$\leq L_h \sum_{x \in \mathcal{X}} \left| \int_{\mathcal{I}} \left( W_N(\frac{i}{N}, \beta) - W(\frac{i}{N}, \beta) \right) \mu_t^\beta(x) \, d\beta \right|$$

$$\leq L_h \sum_{x \in \mathcal{X}} \left| \int_{\mathcal{I}} \left( W_N(\frac{i}{N}, \beta) - N \int_{(\frac{i-1}{N}, \frac{i}{N}]} W(\alpha, \beta) \, d\alpha \right) \mu_t^\beta(x) \, d\beta \right|$$

$$+ L_h \sum_{x \in \mathcal{X}} \left| \int_{\mathcal{I}} \left( N \int_{(\frac{i-1}{N}, \frac{i}{N}]} W(\alpha, \beta) \, d\alpha - W(\frac{i}{N}, \beta) \right) \mu_t^\beta(x) \, d\beta \right|$$

where the latter term can be bounded as

$$L_h \sum_{x \in \mathcal{X}} \left| \int_{\mathcal{I}} \left( N \int_{(\frac{i-1}{N}, \frac{i}{N}]} W(\alpha, \beta) \, d\alpha - W(\frac{i}{N}, \beta) \right) \mu_t^\beta(x) \, d\beta \right|$$

$$\leq L_h \sum_{x \in \mathcal{X}} \left| \int_{\mathcal{I}} N \int_{(\frac{i-1}{N}, \frac{i}{N}]} \left( W(\alpha, \beta) - W(\frac{\lceil N\alpha \rceil}{N}, \beta) \right) \mu_t^\beta(x) \, d\alpha \, d\beta \right|$$

$$\leq L_h |\mathcal{X}| N \cdot \frac{1}{N} \cdot \frac{L_W}{N} = \frac{L_W L_h |\mathcal{X}|}{N}$$

by Assumption 2.

Alternatively, if we assumed the weaker block-wise Lipschitz condition on $W$ in (20), we can obtain the same result for almost all $i \in \mathcal{V}_N$, i.e. for any $p_0 > 0$ there exists $N' \in \mathbb{N}$ such that for any $N > N'$, there exists a set $\mathcal{W}_N^0$, $|\mathcal{W}_N^0| \geq \lfloor (1 - p_0) N \rfloor$ such that for all $i \in \mathcal{W}_N^0$ the above is true: Since by (20) there exist only a finite number $Q$ of intervals and therefore jumps, there can be only $Q$ many $i$ for which the above fails, while for all other $i$ we again have

$$\left| \int_{\mathcal{I}} N \int_{(\frac{i-1}{N}, \frac{i}{N}]} \left( W(\alpha, \beta) - W(\frac{\lceil N\alpha \rceil}{N}, \beta) \right) \mu_t^\beta(x) \, d\alpha \, d\beta \right|$$

$$\leq \sum_{j \in \{1, \dots, Q\}} \left| \int_{\mathcal{I}_j} N \int_{(\frac{i-1}{N}, \frac{i}{N}]} \left( W(\alpha, \beta) - W(\frac{\lceil N\alpha \rceil}{N}, \beta) \right) \mu_t^\beta(x) \, d\alpha \, d\beta \right|$$

$$\leq N \cdot \frac{1}{N} \cdot \frac{L_W}{N} = \frac{L_W L_h |\mathcal{X}|}{N}$$

by (20), as $(\frac{i-1}{N}, \frac{i}{N}] \times \mathcal{I}_j \subseteq \mathcal{I}_k \times \mathcal{I}_j$ for some $k \in \{1, \dots, Q\}$.

For the former term we observe that

$$L_h \sum_{x \in \mathcal{X}} \left| \int_{\mathcal{I}} \left( W_N(\frac{i}{N}, \beta) - N \int_{(\frac{i-1}{N}, \frac{i}{N}]} W(\alpha, \beta) \, d\alpha \right) \mu_t^\beta(x) \, d\beta \right|$$

$$\leq L_h \sum_{x \in \mathcal{X}} N \int_{(\frac{i-1}{N}, \frac{i}{N}]} \left| \int_{\mathcal{I}} (W_N(\alpha, \beta) - W(\alpha, \beta)) \mu_t^\beta(x) \, d\beta \right| d\alpha$$

and by defining for any $x \in \mathcal{X}$ the terms $I_i^N(x)$ via

$$I_i^N(x) := N \int_{(\frac{i-1}{N}, \frac{i}{N}]} \left| \int_{\mathcal{I}} (W_N(\alpha, \beta) - W(\alpha, \beta)) \mu_t^\beta(x) \, d\beta \right| d\alpha$$

and noticing that we have

$$\frac{1}{N} \sum_{i=1}^{N} I_i^N(x) = \int_{\mathcal{I}} \left| \int_{\mathcal{I}} (W_N(\alpha, \beta) - W(\alpha, \beta)) \mu_t^\beta(x) \, d\beta \right| d\alpha \to 0$$

by Assumption 1, we can conclude that for any $\varepsilon_1, p_1 > 0$ there exists $N' \in \mathbb{N}$ such that for any $N > N'$, there exists a set $\mathcal{W}_N^1$, $|\mathcal{W}_N^1| \geq \lfloor (1 - p_1) N \rfloor$ such that for all $i \in \mathcal{W}_N^1$ we have

$$I_i^N(x) < \varepsilon_1,$$

since by the above we can choose $N' \in \mathbb{N}$ such that for any $N > N'$ we have $\frac{1}{N} \sum_{i=1}^{N} I_i^N(x) < \varepsilon_1 p_1$, and from $I_i^N(x) \geq 0$ it would otherwise follow that $\frac{1}{N} \sum_{i=1}^{N} I_i^N(x) \geq \frac{1}{N} \cdot \lceil p_1 N \rceil \varepsilon_1 \geq \varepsilon_1 p_1$ which would be a direct contradiction. Therefore, for all $i \in \mathcal{W}_N^1$, we have uniformly

$$L_h \sum_{x \in \mathcal{X}} N \int_{(\frac{i-1}{N}, \frac{i}{N}]} \left| \int_{\mathcal{I}} (W_N(\alpha, \beta) - W(\alpha, \beta)) \mu_t^\beta(x) \, \mathrm{d}\beta \right| \mathrm{d}\alpha = L_h \sum_{x \in \mathcal{X}} I_i^N(x) \to 0 \,.$$

**Third term.** By (24), for any $\varepsilon_2, p_2 > 0$ there exists a set $\mathcal{W}_N^2$, $|\mathcal{W}_N^2| \geq \lfloor (1 - p_2)N \rfloor$ such that for all $i \in \mathcal{W}_N^2$ we have

$$\left| \mathbb{E}\left[ h(X_t^i, \mathbb{G}^{\frac{i}{N}}(\boldsymbol{\mu}_t)) \right] - \mathbb{E}\left[ h(\hat{X}_t^{\frac{i}{N}}, \mathbb{G}^{\frac{i}{N}}(\boldsymbol{\mu}_t)) \right] \right| < \varepsilon_2$$

independent of $\hat{\pi} \in \Pi$.

The intersection of $\mathcal{W}_N^0$, $\mathcal{W}_N^1$, $\mathcal{W}_N^2$ has at least $N - \lceil p_0 N \rceil - \lceil p_1 N \rceil - \lceil p_2 N \rceil$ agents fulfilling (25), which completes the proof of (24) $\implies$ (25) for almost all agents by choosing $\varepsilon_1, \varepsilon_2$ sufficiently small and $p_0, p_1, p_2 < \frac{p}{3}$ such that $N - \lceil p_0 N \rceil - \lceil p_1 N \rceil - \lceil p_2 N \rceil \geq \lfloor (1-p)N \rfloor$, which is equivalent to $1 - \frac{\lceil p_0 N \rceil}{N} - \frac{\lceil p_1 N \rceil}{N} - \frac{\lceil p_2 N \rceil}{N} \geq \frac{\lfloor (1-p)N \rfloor}{N}$ and is true for sufficiently large $N$, since in the limit, $1 - \frac{\lceil p_0 N \rceil}{N} - \frac{\lceil p_1 N \rceil}{N} - \frac{\lceil p_2 N \rceil}{N} \to 1 - p_0 - p_1 - p_2$ and $\frac{\lfloor (1-p)N \rfloor}{N} \to 1 - p$ as $N \to \infty$.

**Proof of (24).** All that remains is to show (24), which will automatically imply (25) at all times $t \in \mathcal{T}$ by the prequel. We will show (24) by induction.

**Initial case.** At $t = 0$, $\mathcal{L}(X_t^i) = \mu_0 = \mathcal{L}(\hat{X}_t^{\frac{i}{N}})$ by definition. Thus, trivially

$$\left| \mathbb{E}\left[ g(X_0^i) \right] - \mathbb{E}\left[ g(\hat{X}_0^{\frac{i}{N}}) \right] \right| = 0 < \varepsilon \,.$$

**Induction step.** For any uniformly bounded family of functions $\mathcal{G}$ from $\mathcal{X}$ to $\mathbb{R}$ with bound $M_g$, we will show that for any $\varepsilon, p > 0$, there exists $N' \in \mathbb{N}$ such that for all $N > N'$ we have

$$\left| \mathbb{E}\left[ g(X_{t+1}^i) \right] - \mathbb{E}\left[ g(\hat{X}_{t+1}^{\frac{i}{N}}) \right] \right| < \varepsilon$$

uniformly over $\hat{\pi} \in \Pi$, $i \in \mathcal{W}_N$ for some $\mathcal{W}_N \subseteq \mathcal{V}_N$ with $|\mathcal{W}_N| \geq \lfloor (1 - p)N \rfloor$. Observe that

$$\left| \mathbb{E}\left[ g(X_{t+1}^i) \right] - \mathbb{E}\left[ g(\hat{X}_{t+1}^{\frac{i}{N}}) \right] \right| = \left| \mathbb{E}\left[ l_{N,t}(X_t^i, \mathbb{G}_N^{\frac{i}{N}}(\boldsymbol{\mu}_t^N)) \right] - \mathbb{E}\left[ l_{N,t}(\hat{X}_t^{\frac{i}{N}}, \mathbb{G}^{\frac{i}{N}}(\boldsymbol{\mu}_t)) \right] \right|$$

where we defined the uniformly bounded, uniformly Lipschitz functions

$$l_{N,t}(x, \nu) \equiv \sum_{u \in \mathcal{U}} \hat{\pi}_t(u \mid x) \sum_{x' \in \mathcal{X}} P(x' \mid x, u, \nu) g(x')$$

with Lipschitz constant $|\mathcal{X}| M_g L_P$ and uniform bound $M_g$. By the induction assumption and (24) $\implies$ (25) from the prequel, there exists $N' \in \mathbb{N}$ such that for all $N > N'$ we have

$$\left| \mathbb{E}\left[ l_{N,t}(X_t^i, \mathbb{G}_N^{\frac{i}{N}}(\boldsymbol{\mu}_t^N)) \right] - \mathbb{E}\left[ l_{N,t}(\hat{X}_t^{\frac{i}{N}}, \mathbb{G}^{\frac{i}{N}}(\boldsymbol{\mu}_t)) \right] \right| < \varepsilon$$

uniformly over $\hat{\pi} \in \Pi$, $i \in \mathcal{W}_N$ for some $\mathcal{W}_N \subseteq \mathcal{V}_N$ with $|\mathcal{W}_N| \geq \lfloor (1 - p)N \rfloor$, which completes the proof by induction. $\qquad\square$

### A.2.4 PROOF OF COROLLARY A.1

*Proof.* Define the uniformly bounded, uniformly Lipschitz functions

$$r_{\hat{\pi}}(x, \nu) \equiv \sum_{u \in \mathcal{U}} r(x, u, \nu) \hat{\pi}_t(u \mid s)$$

with Lipschitz constant $|U| L_r$ and uniform bound $M_r$ such that by Lemma A.1 and Fubini's theorem, there exists $N' \in \mathbb{N}$ such that for all $N > N'$ we have

$$\left| J_i^N(\pi^1, \ldots, \pi^{i-1}, \hat{\pi}, \pi^{i+1}, \ldots, \hat{\pi}) - J_{\frac{i}{N}}^{\boldsymbol{\mu}}(\hat{\pi}) \right|$$

$$\leq \sum_{t=0}^{T-1} \left| \mathbb{E}\left[ r_{\hat{\pi}_t}(X_t^i, \mathbb{G}^{\frac{i}{N}}(\boldsymbol{\mu}_t)) \right] - \mathbb{E}\left[ r_{\hat{\pi}_t}(\hat{X}_t^{\frac{i}{N}}, \mathbb{G}^{\frac{i}{N}}(\boldsymbol{\mu}_t)) \right] \right| < \varepsilon.$$

uniformly over $\hat{\pi} \in \Pi, i \in \mathcal{W}_N$ for some $\mathcal{W}_N \subseteq \mathcal{V}_N$ with $|\mathcal{W}_N| \geq \lfloor (1-p)N \rfloor$ by choosing the maximum over all $N'$ at each finite time step from Lemma A.1.

In case of the infinite horizon discounted objective, we instead first cut off at a time $T > \frac{\log \frac{\varepsilon(1-\gamma)}{4M_r}}{\log \gamma}$ such that trivially

$$\sum_{t=0}^{T-1} \gamma^t \left| \mathbb{E}\left[ r_{\hat{\pi}_t}(X_t^i, \mathbb{G}^{\frac{i}{N}}(\boldsymbol{\mu}_t)) \right] - \mathbb{E}\left[ r_{\hat{\pi}_t}(\hat{X}_t^{\frac{i}{N}}, \mathbb{G}^{\frac{i}{N}}(\boldsymbol{\mu}_t)) \right] \right|$$

$$+ \gamma^T \sum_{t=T}^{\infty} \gamma^{t-T} \left| \mathbb{E}\left[ r_{\hat{\pi}_t}(X_t^i, \mathbb{G}^{\frac{i}{N}}(\boldsymbol{\mu}_t)) \right] - \mathbb{E}\left[ r_{\hat{\pi}_t}(\hat{X}_t^{\frac{i}{N}}, \mathbb{G}^{\frac{i}{N}}(\boldsymbol{\mu}_t)) \right] \right|$$

$$< \sum_{t=0}^{T-1} \gamma^t \left| \mathbb{E}\left[ r_{\hat{\pi}_t}(X_t^i, \mathbb{G}^{\frac{i}{N}}(\boldsymbol{\mu}_t)) \right] - \mathbb{E}\left[ r_{\hat{\pi}_t}(\hat{X}_t^{\frac{i}{N}}, \mathbb{G}^{\frac{i}{N}}(\boldsymbol{\mu}_t)) \right] \right| + \frac{\varepsilon}{2}$$

and then handle the remaining term analogously to the finite horizon case. $\qquad\square$

### A.2.5  PROOF OF THEOREM 3

*Proof.* By Corollary A.1, for any $\varepsilon > 0$ there exists $N' \in \mathbb{N}$ such that for all $N > N'$ we have

$$\max_{\pi \in \Pi} \left( J_i^N(\pi^1, \ldots, \pi^{i-1}, \pi, \pi^{i+1}, \ldots, \pi^N) - J_i^N(\pi^1, \ldots, \pi^N) \right)$$

$$\leq \max_{\pi \in \Pi} \left( J_i^N(\pi^1, \ldots, \pi^{i-1}, \pi, \pi^{i+1}, \ldots, \pi^N) - J_{\frac{i}{N}}^{\boldsymbol{\mu}}(\pi) \right)$$

$$+ \max_{\pi \in \Pi} \left( J_{\frac{i}{N}}^{\boldsymbol{\mu}}(\pi) - J_{\frac{i}{N}}^{\boldsymbol{\mu}}(\pi^{\frac{i}{N}}) \right)$$

$$+ \left( J_{\frac{i}{N}}^{\boldsymbol{\mu}}(\pi^{\frac{i}{N}}) - J_i^N(\pi^1, \ldots, \pi^N) \right)$$

$$< \frac{\varepsilon}{2} + 0 + \frac{\varepsilon}{2} = \varepsilon$$

uniformly over $i \in \mathcal{W}_N$ for some $\mathcal{W}_N \subseteq \mathcal{V}_N$ with $|\mathcal{W}_N| \geq \lfloor (1-p)N \rfloor$, since $\pi^{\frac{i}{N}} \in \arg\max_\pi J_{\frac{i}{N}}^{\boldsymbol{\mu}}(\pi)$ by definition of a GMFE. Reordering completes the proof. $\qquad\square$

### A.2.6  PROOF OF COROLLARY A.2

*Proof.* The proof follows immediately from Theorem 2 and Lemma A.1 by considering the trivial policy $\boldsymbol{\pi}$ that always chooses the only action available together with its generated mean field $\boldsymbol{\mu} = \Psi(\boldsymbol{\pi})$. $\qquad\square$

### A.2.7  PROOF OF PROPOSITION 2

*Proof.* The set $\boldsymbol{\Pi}$ is a complete metric space, since existence of limits follows from completeness of $\mathbb{R}$, pointwise limits of measurable functions are measurable, and policies will remain normalized. Banach's fixed point theorem applied to $\hat{\Phi} \circ \hat{\Psi}$ gives us the desired result. $\qquad\square$

### A.2.8  PROOF OF THEOREM 4

*Proof.* Formally, we approximate mean fields by $\hat{\Psi}(\boldsymbol{\pi}) = \sum_{i=1}^M \mathbf{1}_{\alpha \in \tilde{\mathcal{I}}_i} \hat{\mu}^{\alpha_i}$ for any fixed policy ensemble $\boldsymbol{\pi}$, and similarly policies $\hat{\Phi}(\boldsymbol{\mu}) = \sum_{i=1}^M \mathbf{1}_{\alpha \in \tilde{\mathcal{I}}_i} \pi^{\alpha_i}$ where $\pi^{\alpha_i}$ is the softmax policy of $\alpha_i$ for fixed $\boldsymbol{\mu}$, i.e.

$$\pi_t^\alpha(u \mid x) = \frac{\exp\left( \frac{Q_\alpha^{\boldsymbol{\mu}}(t,x,u)}{\eta} \right)}{\sum_{u \in \mathcal{U}} \exp\left( \frac{Q_\alpha^{\boldsymbol{\mu}}(t,x,u)}{\eta} \right)} . \tag{30}$$

By the Bellman equation (13), $Q^{\boldsymbol{\mu}}_\alpha(t, x, u)$ is Lipschitz in $\boldsymbol{\mu}$ for all $(t, x, u) \in \mathcal{T} \times \mathcal{X} \times \mathcal{U}$ under Assumption 2. Since the Lipschitz constants are shared over all $\alpha$, by Cui & Koeppl (2021), Lemma B.7.5, (30) is therefore Lipschitz with Lipschitz constant proportional to $1/\eta$, which immediately implies that $\hat{\Phi}$ is also Lipschitz with Lipschitz constant $c_1/\eta$. By its recursive definition as compositions of Lipschitz functions, $\hat{\Psi}$ is Lipschitz as well with some constant $c_2$. Therefore, the composition of both functions $\hat{\Psi} \circ \hat{\Phi}$ is Lipschitz with constants $c_1 c_2/\eta$, which will be less than 1 for sufficiently large $\eta$. By Proposition 2, the equivalence classes algorithm $\hat{\Psi} \circ \hat{\Phi}$ converges to a fixed point. $\qquad \square$

### A.2.9 PROOF OF THEOREM 5

*Proof.* First, note that under the equivalence classes method, the distance between any $\alpha$ and its representant $\alpha_i$ uniformly shrinks to zero as $M \to \infty$, i.e. $\max_{i=1,\dots,M} \sup_{\alpha \in \tilde{\mathcal{I}}_i} |\alpha - \alpha_i| \to 0$.

We begin by showing that a solution of the $M$ equivalence classes method $(\boldsymbol{\pi}, \boldsymbol{\mu}) \in \boldsymbol{\Pi} \times \boldsymbol{\mathcal{M}}$, $\boldsymbol{\pi} \in \hat{\Phi}(\boldsymbol{\mu})$, $\boldsymbol{\mu} = \hat{\Psi}(\boldsymbol{\pi})$ following (31), (32) fulfills approximate optimality, i.e. for any $\varepsilon > 0$ there exists $M'$ s.t. for all $M > M'$

$$\sup_{\alpha \in \mathcal{I}} \max_{\pi \in \Pi} \left( J^{\bar{\boldsymbol{\mu}}}_\alpha(\pi) - J^{\bar{\boldsymbol{\mu}}}_\alpha(\pi^\alpha) \right) < \varepsilon,$$

where we introduced the true, exact mean field ensemble $\bar{\boldsymbol{\mu}} = \Psi(\boldsymbol{\pi})$ following (12) generated by the block-wise solution policy $\sum_{i=1}^M \mathbf{1}_{\alpha \in \tilde{\mathcal{I}}_i} \pi^{\alpha_i}$ of the $M$ equivalence classes method, as well as the true mean field system under $\bar{\boldsymbol{\mu}}$ and any policy $\pi \in \Pi$

$$\bar{X}^\alpha_0 \sim \mu_0, \quad \bar{U}^\alpha_t \sim \pi_t(\cdot \mid \bar{X}^\alpha_t), \quad \bar{X}^\alpha_{t+1} \sim P(\cdot \mid \bar{X}^\alpha_t, \bar{U}^\alpha_t, \bar{\mathbb{G}}^\alpha_t), \quad \forall (\alpha, t) \in \mathcal{I} \times \mathcal{T}$$

with $\mathcal{B}_1(\mathcal{X})$-valued $\bar{\mathbb{G}}^\alpha_t := \int_\mathcal{I} W(\alpha, \beta) \bar{\mu}^\beta_t \, \mathrm{d}\beta$ and $J^{\bar{\boldsymbol{\mu}}}_\alpha(\pi) \equiv \mathbb{E}\left[ \sum_{t=0}^{T-1} r(\bar{X}^\alpha_t, \bar{U}^\alpha_t, \bar{\mathbb{G}}^\alpha_t) \right]$, while system (9) is to be understood as the system under the approximate mean field ensemble $\boldsymbol{\mu}$.

To see this, we will analyze

$$
\begin{aligned}
\sup_{\alpha \in \mathcal{I}} \max_{\pi \in \Pi} \left( J^{\bar{\boldsymbol{\mu}}}_\alpha(\pi) - J^{\bar{\boldsymbol{\mu}}}_\alpha(\pi^\alpha) \right) \leq\ & \max_{i=1,\dots,M} \sup_{\alpha \in \tilde{\mathcal{I}}_i} \max_{\pi \in \Pi} \left( J^{\bar{\boldsymbol{\mu}}}_\alpha(\pi) - J^{\boldsymbol{\mu}}_\alpha(\pi) \right) \\
& + \max_{i=1,\dots,M} \sup_{\alpha \in \tilde{\mathcal{I}}_i} \max_{\pi \in \Pi} \left( J^{\boldsymbol{\mu}}_\alpha(\pi) - J^{\boldsymbol{\mu}}_{\alpha_i}(\pi) \right) \\
& + \max_{i=1,\dots,M} \sup_{\alpha \in \tilde{\mathcal{I}}_i} \max_{\pi \in \Pi} \left( J^{\boldsymbol{\mu}}_{\alpha_i}(\pi) - J^{\boldsymbol{\mu}}_{\alpha_i}(\pi^{\alpha_i}) \right) \\
& + \max_{i=1,\dots,M} \sup_{\alpha \in \tilde{\mathcal{I}}_i} \left( J^{\boldsymbol{\mu}}_{\alpha_i}(\pi^{\alpha_i}) - J^{\boldsymbol{\mu}}_\alpha(\pi^{\alpha_i}) \right) \\
& + \max_{i=1,\dots,M} \sup_{\alpha \in \tilde{\mathcal{I}}_i} \left( J^{\boldsymbol{\mu}}_\alpha(\pi^\alpha) - J^{\bar{\boldsymbol{\mu}}}_\alpha(\pi^\alpha) \right).
\end{aligned}
$$

**First term.** For any $\pi \in \Pi$, define the uniformly bounded, uniformly Lipschitz functions

$$r_\pi(x, \nu) \equiv \sum_{u \in \mathcal{U}} r(x, u, \nu) \pi_t(u \mid s)$$

with Lipschitz constant $|U| L_r$ and uniform bound $M_r$ such that for the first term, we have

$$
\begin{aligned}
\left( J^{\bar{\boldsymbol{\mu}}}_\alpha(\pi) - J^{\boldsymbol{\mu}}_\alpha(\pi) \right) &\leq \left| J^{\bar{\boldsymbol{\mu}}}_\alpha(\pi) - J^{\boldsymbol{\mu}}_\alpha(\pi) \right| \\
&\leq \sum_{t=0}^{T-1} \left| \mathbb{E}\left[ r_\pi(\bar{X}^\alpha_t, \bar{\mathbb{G}}^\alpha_t) \right] - \mathbb{E}\left[ r_\pi(X^\alpha_t, \mathbb{G}^\alpha_t) \right] \right| \\
&\leq \sum_{t=0}^{T-1} \left| \mathbb{E}\left[ r_\pi(\bar{X}^\alpha_t, \bar{\mathbb{G}}^\alpha_t) - r_\pi(\bar{X}^\alpha_t, \mathbb{G}^\alpha_t) \right] \right| + \sum_{t=0}^{T-1} \left| \mathbb{E}\left[ r_\pi(\bar{X}^\alpha_t, \mathbb{G}^\alpha_t) \right] - \mathbb{E}\left[ r_\pi(X^\alpha_t, \mathbb{G}^\alpha_t) \right] \right| \\
&\leq \sum_{t=0}^{T-1} |U| L_r \left\| \int_\mathcal{I} W(\alpha, \beta)(\bar{\mu}^\beta_t - \mu^\beta_t) \, \mathrm{d}\beta \right\| + \sum_{t=0}^{T-1} \left| \mathbb{E}\left[ r_\pi(\bar{X}^\alpha_t, \mathbb{G}^\alpha_t) \right] - \mathbb{E}\left[ r_\pi(X^\alpha_t, \mathbb{G}^\alpha_t) \right] \right|.
\end{aligned}
$$

For the former term, note that

$$\left\| \int_{\mathcal{I}} W(\alpha, \beta)(\bar{\mu}_t^\beta - \mu_t^\beta) \, \mathrm{d}\beta \right\| = \left\| \sum_i \int_{\tilde{\mathcal{I}}_i} W(\alpha, \beta)(\bar{\mu}_t^\beta - \mu_t^{\alpha_i}) \, \mathrm{d}\beta \right\|$$
$$\leq \max_{i=1,\ldots,M} \sup_{\alpha \in \tilde{\mathcal{I}}_i} |\mathcal{X}| \left\| \bar{\mu}_t^\alpha - \mu_t^{\alpha_i} \right\|$$

and we will show by induction over $t = 0, 1, \ldots, T$ that $\sup_{\alpha \in \tilde{\mathcal{I}}_i} \|\bar{\mu}_t^\alpha - \mu_t^{\alpha_i}\| \to 0$ over all $\alpha$ uniformly over all equivalence classes $\tilde{\mathcal{I}}_i$. At $t = 0$, we have trivially $\bar{\mu}_0 = \mu_0$. Assume that $\sup_{\alpha \in \tilde{\mathcal{I}}_i} \|\bar{\mu}_t^\alpha - \mu_t^{\alpha_i}\| \to 0$. Then for $t+1$, we have

$$\sup_{\alpha \in \tilde{\mathcal{I}}_i} \left\| \bar{\mu}_{t+1}^\alpha - \mu_{t+1}^{\alpha_i} \right\|$$

$$= \sup_{\alpha \in \tilde{\mathcal{I}}_i} \left\| \sum_{x \in \mathcal{X}} \bar{\mu}_t^\alpha(x) \sum_{u \in \mathcal{U}} \pi_t^\alpha(u \mid x) P(\cdot \mid x, u, \bar{\mathbb{G}}_t^\alpha) - \sum_{x \in \mathcal{X}} \mu_t^{\alpha_i}(x) \sum_{u \in \mathcal{U}} \pi_t^{\alpha_i}(u \mid x) P(\cdot \mid x, u, \mathbb{G}_t^{\alpha_i}) \right\|$$

$$\leq \sup_{\alpha \in \tilde{\mathcal{I}}_i} \left\| \sum_{x \in \mathcal{X}} \bar{\mu}_t^\alpha(x) \sum_{u \in \mathcal{U}} \pi_t^\alpha(u \mid x) P(\cdot \mid x, u, \bar{\mathbb{G}}_t^\alpha) - \sum_{x \in \mathcal{X}} \bar{\mu}_t^{\alpha_i}(x) \sum_{u \in \mathcal{U}} \pi_t^{\alpha_i}(u \mid x) P(\cdot \mid x, u, \bar{\mathbb{G}}_t^{\alpha_i}) \right\|$$

$$+ \left\| \sum_{x \in \mathcal{X}} \bar{\mu}_t^{\alpha_i}(x) \sum_{u \in \mathcal{U}} \pi_t^{\alpha_i}(u \mid x) P(\cdot \mid x, u, \bar{\mathbb{G}}_t^{\alpha_i}) - \sum_{x \in \mathcal{X}} \mu_t^{\alpha_i}(x) \sum_{u \in \mathcal{U}} \pi_t^{\alpha_i}(u \mid x) P(\cdot \mid x, u, \bar{\mathbb{G}}_t^{\alpha_i}) \right\|$$

$$+ \left\| \sum_{x \in \mathcal{X}} \bar{\mu}_t^{\alpha_i}(x) \sum_{u \in \mathcal{U}} \pi_t^{\alpha_i}(u \mid x) P(\cdot \mid x, u, \bar{\mathbb{G}}_t^{\alpha_i}) - \sum_{x \in \mathcal{X}} \mu_t^{\alpha_i}(x) \sum_{u \in \mathcal{U}} \pi_t^{\alpha_i}(u \mid x) P(\cdot \mid x, u, \mathbb{G}_t^{\alpha_i}) \right\|$$

$$\leq \sup_{\alpha \in \tilde{\mathcal{I}}_i} \left\| \sum_{x \in \mathcal{X}} \bar{\mu}_t^\alpha(x) \sum_{u \in \mathcal{U}} \pi_t^\alpha(u \mid x) P(\cdot \mid x, u, \bar{\mathbb{G}}_t^\alpha) - \sum_{x \in \mathcal{X}} \bar{\mu}_t^{\alpha_i}(x) \sum_{u \in \mathcal{U}} \pi_t^{\alpha_i}(u \mid x) P(\cdot \mid x, u, \bar{\mathbb{G}}_t^{\alpha_i}) \right\|$$
$$+ |\mathcal{X}|^2 \left\| \bar{\mu}_t^{\alpha_i} - \mu_t^{\alpha_i} \right\| + |\mathcal{X}|^2 |\mathcal{U}| L_P \left\| \bar{\mu}_t^{\alpha_i} - \mu_t^{\alpha_i} \right\| \to 0$$

as $M \to \infty$, since the first term is uniformly Lipschitz in $\alpha$ by (12) as a recursive composition, finite multiplication and addition of Lipschitz functions, whereas the other terms tend to zero by induction hypothesis. Since the Lipschitz constants do not depend on $\tilde{\mathcal{I}}_i$, the convergence is uniform.

To bound the latter term, we first note that $r_\pi(\cdot, \mathbb{G}_t^\alpha)$ is always bounded by $M_r$ regardless of $t, \alpha, \pi$, i.e. it again suffices to show that for any family of functions $\mathcal{G}$ from $\mathcal{X}$ to $\mathbb{R}$ uniformly bounded by $M_r$, we have

$$\sup_{g \in \mathcal{G}} \left| \mathbb{E}\left[ g(\bar{X}_t^\alpha) \right] - \mathbb{E}\left[ g(X_t^\alpha) \right] \right| \to 0 \,.$$

The proof is by induction. At $t = 0$, we trivially have $\mathcal{L}(\bar{X}_t^\alpha) = \mu_0 = \mathcal{L}(X_t^\alpha)$. Assuming that the induction hypothesis holds at $t$, then at $t+1$ we have

$$\sup_{g \in \mathcal{G}} \left| \mathbb{E}\left[ g(\bar{X}_{t+1}^\alpha) \right] - \mathbb{E}\left[ g(X_{t+1}^\alpha) \right] \right| = \sup_{g \in \mathcal{G}} \left| \mathbb{E}\left[ l_t(\bar{X}_t^\alpha, \mathbb{G}_t^\alpha) \right] - \mathbb{E}\left[ l_t(X_t^\alpha, \mathbb{G}_t^\alpha) \right] \right| \to 0$$

by the induction hypothesis, where we defined the uniformly bounded functions

$$l_t(x, \nu) \equiv \sum_{u \in \mathcal{U}} \pi_t(u \mid x) \sum_{x' \in \mathcal{X}} P(x' \mid x, u, \nu) g(x')$$

with uniform bound $M_r$. Therefore, $|J_\alpha^{\bar{\mu}}(\pi) - J_\alpha^{\mu}(\pi)| \to 0$ uniformly over all $\alpha, \pi$.

**Second term.** For the second term, we analogously have

$$\left( J_\alpha^{\mu}(\pi) - J_{\alpha_i}^{\mu}(\pi) \right) \leq \left| J_\alpha^{\mu}(\pi) - J_\alpha^{\mu}(\pi) \right|$$
$$\leq \sum_{t=0}^{T-1} |U| L_r \left\| \int_{\mathcal{I}} (W(\alpha, \beta) - W(\alpha_i, \beta)) \mu_t^\beta \, \mathrm{d}\beta \right\| + \sum_{t=0}^{T-1} \left| \mathbb{E}\left[ r_\pi(X_t^\alpha, \mathbb{G}_t^{\alpha_i}) \right] - \mathbb{E}\left[ r_\pi(X_t^{\alpha_i}, \mathbb{G}_t^{\alpha_i}) \right] \right|$$

where the former term uniformly tends to zero as $M \to \infty$ over all $\alpha$ by Lipschitz $W$ from Assumption 2 and increasingly fine partition intervals $\tilde{\mathcal{I}}_i$, while for the latter term we again show that for any family of functions $\mathcal{G}$ from $\mathcal{X}$ to $\mathbb{R}$ uniformly bounded by $M_r$, we have

$$\sup_{g \in \mathcal{G}} |\mathbb{E}[g(X_t^\alpha)] - \mathbb{E}[g(X_t^{\alpha_i})]| \to 0.$$

The proof is by induction. At $t = 0$, we trivially have $\mathcal{L}(X_0^\alpha) = \mu_0 = \mathcal{L}(X_0^{\alpha_i})$. Assuming that the induction hypothesis holds at $t$, then at $t + 1$ we have

$$\sup_{g \in \mathcal{G}} \left|\mathbb{E}\left[g(X_{t+1}^\alpha)\right] - \mathbb{E}\left[g(X_{t+1}^{\alpha_i})\right]\right| = \sup_{g \in \mathcal{G}} |\mathbb{E}[l_t(X_t^\alpha, \mathbb{G}_t^\alpha)] - \mathbb{E}[l_t(X_t^{\alpha_i}, \mathbb{G}_t^{\alpha_i})]| \to 0$$

by the induction hypothesis, where we defined the uniformly bounded functions

$$l_t(x, \nu) \equiv \sum_{u \in \mathcal{U}} \pi_t(u \mid x) \sum_{x' \in \mathcal{X}} P(x' \mid x, u, \nu)g(x')$$

with uniform bound $M_r$. Therefore, $\left|J_\alpha^{\boldsymbol{\mu}}(\pi) - J_{\alpha_i}^{\boldsymbol{\mu}}(\pi)\right| \to 0$ uniformly over all $\alpha, \pi$.

**Third term.** By definition, we have optimality of $\boldsymbol{\pi} \in \hat{\Phi}(\boldsymbol{\mu})$ under the approximate mean field $\boldsymbol{\mu}$ at each representative $\alpha_i$. Therefore, the term $\max_{\pi \in \Pi} \left(J_{\alpha_i}^{\boldsymbol{\mu}}(\pi) - J_{\alpha_i}^{\boldsymbol{\mu}}(\pi^{\alpha_i})\right)$ is upper bounded by 0, as there is no policy $\pi$ that improves over $\pi^{\alpha_i}$.

**Fourth and fifth term.** The results follow from the first and second term by inserting $\pi^\alpha$ for $\pi$.

**Variations on the setting.** The infinite horizon discounted case is handled as in the proof of Corollary A.1, i.e. repeating the above up to some chosen time horizon $T$ and trivially bounding all terms with $t \geq T$. The block-wise Lipschitz graphon case (20) is handled by choosing the equivalence classes $\tilde{\mathcal{I}}_i \subseteq \mathcal{I}_j$ such that they are part of at most one block $\mathcal{I}_j$ of the graphon.

**Proof of Theorem 5.** Now fix any $\varepsilon, p > 0$. As a result of the prequel, we have that there exists $M'$ s.t. for all $M > M'$

$$\sup_{\alpha \in \mathcal{I}} \max_{\pi \in \Pi} |J_\alpha^{\boldsymbol{\mu}}(\pi^\alpha) - J_\alpha^{\boldsymbol{\mu}}(\pi)| < \frac{\varepsilon}{3}.$$

Pick any such $M > M'$. By Corollary A.1 (for the first and third term, since $\boldsymbol{\pi}$ is constant with at most $M$ discontinuities) and the prequel (for the second term), there exists $N' \in \mathbb{N}$ such that for all $N > N'$ we have

$$\max_{\pi \in \Pi} \left(J_i^N(\pi^1, \ldots, \pi^{i-1}, \pi, \pi^{i+1}, \ldots, \pi^N) - J_i^N(\pi^1, \ldots, \pi^N)\right)$$

$$\leq \max_{\pi \in \Pi} \left(J_i^N(\pi^1, \ldots, \pi^{i-1}, \pi, \pi^{i+1}, \ldots, \pi^N) - J_{\frac{i}{N}}^{\bar{\boldsymbol{\mu}}}(\pi)\right)$$

$$+ \max_{\pi \in \Pi} \left(J_{\frac{i}{N}}^{\bar{\boldsymbol{\mu}}}(\pi) - J_{\frac{i}{N}}^{\bar{\boldsymbol{\mu}}}(\pi^{\frac{i}{N}})\right)$$

$$+ \left(J_{\frac{i}{N}}^{\bar{\boldsymbol{\mu}}}(\pi^{\frac{i}{N}}) - J_i^N(\pi^1, \ldots, \pi^N)\right)$$

$$< \frac{\varepsilon}{3} + \frac{\varepsilon}{3} + \frac{\varepsilon}{3} = \varepsilon$$

which holds uniformly over $i \in \mathcal{W}_N$ for some $\mathcal{W}_N \subseteq \mathcal{V}_N$ with $|\mathcal{W}_N| \geq \lfloor (1 - p)N \rfloor$, since $\pi^{\frac{i}{N}} \in \arg\max_\pi J_{\frac{i}{N}}^{\boldsymbol{\mu}}(\pi)$ by definition of a GMFE. Reordering completes the proof. $\square$

## A.3 EXPERIMENTAL DETAILS

In this section, we will give a full description of all the algorithms and hyperparameters we used during our experiments. For reinforcement learning, we use PPO (Schulman et al., 2017).

For the approximate equivalence classes, we shall consider grids $(\alpha_m \in [0, 1])_{m=1,\ldots,M}$ with associated policies $(\pi^{\alpha_m} \in \Pi)_{m=1,\ldots,M}$ and mean fields $(\mu^{\alpha_m} \in \mathcal{P}(\mathcal{X})^\mathcal{T})_{m=1,\ldots,M}$. For the grid, we

---

**Algorithm 1 Fixed point iteration**

---

1: Initialize $\boldsymbol{\mu}^0$ as the mean field induced by the uniformly random policy $\mathbf{q}$.
2: **for** $k = 0, 1, \ldots$ **do**
3:     Compute $\boldsymbol{\pi}^k \in \boldsymbol{\Pi}$ either directly by PPO on (14), or by computing $\mathbf{Q}^{\boldsymbol{\mu}}$ via Algorithm 2 and using (38) to obtain a softmax policy.
4:     Compute $\boldsymbol{\mu}^{k+1}$ induced by $\pi^k$ using Algorithm 3, or for RL the neighborhood mean fields $\mathbb{G}_t^{\alpha}$ directly using Algorithm 4.
5: **end for**

---

**Algorithm 2 Backwards induction**

---

1: **Input**: Grid $(\alpha_m \in [0, 1])_{m=1,\ldots,M}$, mean field $\boldsymbol{\mu} \in \mathcal{M}$.
2: **for** $m = 1, \ldots, M$ **do**
3:     Initialize terminal condition $Q_\alpha^{\boldsymbol{\mu}}(T, x, u) \equiv 0$ for all $(x, u) \in \mathcal{X} \times \mathcal{U}$.
4:     **for** $t = T - 1, \ldots, 0$ **do**
5:         **for** $(x, u) \in \mathcal{X} \times \mathcal{U}$ **do**
6:             $Q_{\alpha_m}^{\boldsymbol{\mu}}(t, x, u) \leftarrow r(x, u, \mathbb{G}_t^{\alpha_m}) + \sum_{x' \in \mathcal{X}} P(x' \mid x, u, \mathbb{G}_t^{\alpha_m}) \max_{u' \in \mathcal{U}} Q_{\alpha_m}^{\boldsymbol{\mu}}(t+1, x', u')$.
7:         **end for**
8:     **end for**
9: **end for**
10: Return $(Q_{\alpha_m}^{\boldsymbol{\mu}})_{m=1,\ldots,M}$

---

choose the points $\alpha_m = \frac{m}{100}$ with $m = 0, \ldots, 100$. Here, an agent $\alpha$ shall use the policy $\pi^{\alpha_m}$ with the closest $\alpha_m$.

To be precise, for the approximate mean field $\boldsymbol{\mu} = \hat{\Psi}(\boldsymbol{\pi})$ we define $\mu^\alpha \equiv \hat{\mu}^{\alpha_m}$ for the $\alpha_m$ closest to $\alpha$, i.e. formally, we thus have

$$\hat{\Psi}(\boldsymbol{\pi}) = \sum_{m=1}^{M} \mathbf{1}_{\alpha \in \tilde{\mathcal{I}}_m} \hat{\mu}^{\alpha_m} \tag{31}$$

for any fixed policy ensemble $\boldsymbol{\pi}$, with $\hat{\mu}$ defined through the recursive equation

$$\hat{\mu}_0^{\alpha_m} \equiv \mu_0, \quad \hat{\mu}_{t+1}^{\alpha_m}(x') \equiv \sum_{x \in \mathcal{X}} \hat{\mu}_t^{\alpha_m}(x) \sum_{u \in \mathcal{U}} \pi_t^{\alpha_m}(u \mid x) P(x' \mid x, u, \hat{\mathbb{G}}_t^{\alpha_m}), \quad m = 1, \ldots, M \tag{32}$$

where under the assumption of equivalence classes $\tilde{\mathcal{I}}_m \equiv [a_m, b_m]$ of size $(b_m - a_m)$, we obtain neighborhood mean fields via

$$\hat{\mathbb{G}}_t^{\alpha} = \sum_{m=1}^{M} (b_m - a_m) W(\alpha, \alpha_m) \hat{\mu}_t^{\alpha_m} . \tag{33}$$

Note that in our algorithms, we shall assume equisized partitions and use $(b_m - a_m) = \frac{1}{M}$. Similarly, the policy ensemble is approximated by

$$\hat{\Phi}(\boldsymbol{\mu}) = \sum_{i=1}^{M} \mathbf{1}_{\alpha \in \tilde{\mathcal{I}}_i} \pi^{\alpha_i} \tag{34}$$

where $\pi^{\alpha_i}$ is the optimal policy of $\alpha_i$ for any fixed $\boldsymbol{\mu}$, i.e. the optimal policy and mean field of each $\alpha$ is approximated by the optimal solution and mean field of the closest $\alpha_i$, which is an increasingly good approximation for sufficiently fine grids under the standing Lipschitz assumptions. In the case of block-wise Lipschitz continuous graphons via (20), a similar justification holds as long as each equivalence class remains constrained to one of the blocks of the graphon, see Theorem 5.

In Algorithm 1, the learning scheme is described on a high level. In our experiments, we either use approximate equivalence classes via Algorithms 2 and 3, or reinforcement learning in the form of PPO together with sequential Monte Carlo in Algorithm 4, though in principle one can mix arbitrary methods.

---

**Algorithm 3 Forward simulation**

---

1: **Input**: Grid $(\alpha_m \in [0,1])_{m=1,\dots,M}$, policy $\boldsymbol{\pi} \in \boldsymbol{\Pi}$.
2: Initialize starting condition $\mu_0^{\alpha_m} \equiv \mu_0$ for all $m = 1, \dots, M$.
3: **for** $t = 0, \dots, T-2$ **do**
4:     **for** $m = 1, \dots, M$ **do**
5:       $\mu_{t+1}^{\alpha_m} \leftarrow \sum_{x \in \mathcal{X}} \mu_t^{\alpha_m}(x) \sum_{u \in \mathcal{U}} \pi_t^{\alpha_m}(u \mid x) P(\cdot \mid x, u, \frac{1}{M} \sum_{n=1}^{M} W(\alpha_m, \alpha_n) \mu_t^{\alpha_n})$.
6:     **end for**
7: **end for**
8: Return $(\mu^{\alpha_m})_{m=1,\dots,M}$

---

**Algorithm 4 Sequential Monte Carlo**

---

1: **Input**: Number of trajectories $K = 5$, number of particles $L = 200$, policy $\boldsymbol{\pi} \in \boldsymbol{\Pi}$.
2: **for** $k = 1, \dots, K$ **do**
3:     Initialize particles $\alpha_m \sim \text{Unif}([0,1])$, $x_0^{m,k} \sim \mu_0$ for all $m = 1, \dots, L$.
4:     **for** $t = 1, \dots, T-1$ **do**
5:       **for** $m = 1, \dots, L$ **do**
6:         Sample action $u \sim \pi_t^{\alpha_m}(\cdot \mid x_t^{m,k})$.
7:         Sample new particle state $x_{t+1}^{m,k} \sim P(\cdot \mid x_t^{m,k}, u, \frac{1}{L} \sum_{n=1}^{L} W(\alpha_m, \alpha_n) \delta_{x_t^{n,k}})$.
8:       **end for**
9:     **end for**
10: **end for**
11: **return** neighborhood mean fields $\mathbb{G}_t^\alpha \approx \frac{1}{K} \sum_{k=1}^{K} \frac{1}{L} \sum_{m=1}^{L} W(\alpha, \alpha_m) \delta_{x_t^{m,k}}$.

---

We ran each trial of our experiments on a single conventional CPU core, with typical wall-clock times reaching up to at most a few days. We estimate the required compute to approximately 6500 core hours. We did not use any GPUs or TPUs. More specifically, the training of our approximate equivalence class approach took on average approximately 24 hours for 250 iterations in SIS and 50 iterations in Investment. As a result, Figure 5 for the selection of appropriate temperatures took around 2500 core hours. The PPO experiments took approximately 3 days for each configuration, resulting in approximately 200 core hours for Figure 7. Finally, for the $N$-agent evaluations in Figure 4, each run up to 100 agents takes up to 4 core hours. Adding on top of that around 250 core hours for the rest of the experiments results in a total of approximately 4000 core hours.

For PPO, we used the RLlib implementation by Liang et al. (2018) (version 1.2.0, Apache-2.0 license). To allow for time-dependent policies, we append the current time to the network inputs. Further, discrete-valued observations are one-hot encoded. Any other parameter configurations are given in Algorithms 1, 2, 3 and 4, as well as in Table 2.

As for the specific configurations used in the PPO experiments, we give the hyperparameters in Table 1 and used with a feedforward neural network policy consisting of two hidden layers with 256 nodes and `tanh` activations, outputting a softmax policy over all actions.

### A.3.1 PROBLEM DEFINITIONS

For each possible problem setting, we list the applied temperature setting in Table 2. In the following, let $\mathbb{G} \in \mathcal{P}(\mathcal{X})$.

**SIS-Graphon.** In the SIS-Graphon game as described in the main text, we have $\mathcal{X} = \{S, I\}$, $\mathcal{U} = \{U, D\}$, $\mu_0(I) = 0.5$, $r(x, u, \mathbb{G}) = -2 \cdot \mathbf{1}_{\{I\}}(x) - 0.5 \cdot \mathbf{1}_{\{D\}}(u)$ and $\mathcal{T} = \{0, \dots, 49\}$. Similar parameters produce similar results, and we set the transition probabilities as

$$\mathbb{P}(S \mid I, \cdot, \cdot) = 0.2,$$
$$\mathbb{P}(I \mid S, U, \mathbb{G}) = 0.8 \cdot \mathbb{G}(I),$$
$$\mathbb{P}(I \mid S, D, \cdot) = 0.$$

**Investment-Graphon.** Similarly, in the Investment-Graphon game we have $\mathcal{X} = \{0, 1, \dots 9\}$, $\mathcal{U} = \{I, O\}$, $\mu_0(0) = 1$, $r(x, u, \mathbb{G}) = \frac{0.3x}{1 + \sum_{x' \in \mathcal{X}} x' \mathbb{G}(x')} - 2 \cdot \mathbf{1}_{\{I\}}(u)$ and $\mathcal{T} = \{0, \dots, 49\}$. We set

Table 1: PPO Hyperparameters

| Symbol | Function | Value |
|---|---|---|
| $l_r$ | Learning rate | 0.00005 |
| $\gamma$ | Discount rate | 1 |
| $\lambda$ | GAE lambda | 0.99 |
| $c_{\mathrm{KL}}$ | KL coefficient | 0.2 |
| $\beta$ | KL target | 0.006 |
| $c_{\mathrm{ent}}$ | Entropy coefficient | 0.01 |
| $\epsilon$ | Clip parameter | 0.2 |
| $B$ | Training batch size | 4000 |
| $B_m$ | Mini batch size | 128 |
| $I_{\mathrm{SGD}}$ | SGD iterations per training batch | 30 |

Table 2: Temperature configurations

| Experiment | $\eta$ for approximate equivalence classes |
|---|---|
| SIS-Graphon, $W_{\mathrm{unif}}$ | 0.101 |
| SIS-Graphon, $W_{\mathrm{rank}}$ | 0.3 |
| SIS-Graphon, $W_{\mathrm{er}}$ | 0.101 |
| Investment-Graphon, $W_{\mathrm{unif}}$ | 0 |
| Investment-Graphon, $W_{\mathrm{rank}}$ | 0 |
| Investment-Graphon, $W_{\mathrm{er}}$ | 0.05 |

the transition probabilities for $x = 0, 1, \ldots, 8$ as

$$\mathbb{P}(x + 1 \mid x, I, \cdot) = \frac{9 - x}{10},$$

$$\mathbb{P}(x \mid x, I, \cdot) = \frac{1 + x}{10},$$

$$\mathbb{P}(x \mid x, O, \cdot) = 1,$$

while for $x = 9$ the next state is always $x = 9$.

### A.3.2  EXPLOITABILITY AND TEMPERATURE CHOICE

In the following, we will explain our choice of temperatures in Table 2 by approximately evaluating the average exploitability of GMFE candidates $(\boldsymbol{\pi}, \boldsymbol{\mu})$ – as it is intractable to approximately evaluate the maximum exploitability over all $\alpha \in \mathcal{I}$ – defined by

$$\Delta J(\boldsymbol{\pi}, \boldsymbol{\mu}) = \int_{\mathcal{I}} \sup_{\pi^* \in \Pi} J_\alpha^{\boldsymbol{\mu}}(\pi^*) - J_\alpha^{\boldsymbol{\mu}}(\pi^\alpha) \, \mathrm{d}\alpha \,. \tag{35}$$

More specifically, when using approximate equivalence classes, we compute the exploitability of some policy $\boldsymbol{\pi}$ by computing the optimal policy $\boldsymbol{\pi}^*$ obtained via Algorithm 2, under the fixed mean field $\boldsymbol{\mu}$ generated by $\boldsymbol{\pi}$ via Algorithm 3, inserting $\pi^{*,\alpha}$ into (35) and then approximating by

$$\int_{\mathcal{I}} J_\alpha^{\boldsymbol{\mu}}(\pi) \, \mathrm{d}\alpha \approx \frac{1}{M} \sum_{m=1,\ldots,M} \sum_{x \in \mathcal{X}} \mu_0(x) \sum_{u \in \mathcal{U}} \pi_0(u \mid x) Q_{\alpha_m}^{\boldsymbol{\mu},\pi}(0, x, u) \,. \tag{36}$$

Here, we defined for any policy $\pi \in \Pi$ and $\alpha \in \mathcal{I}$ the policy evaluation functions $Q_\alpha^{\boldsymbol{\mu},\pi}$ as usual via

$$Q_\alpha^{\boldsymbol{\mu},\pi}(t, x, u) = r(x, u, \mathbb{G}_t^\alpha) + \sum_{x' \in \mathcal{X}} P(x' \mid x, u, \mathbb{G}_t^\alpha) \sum_{u' \in \mathcal{U}} \pi_0(u' \mid x) Q_\alpha^{\boldsymbol{\mu},\pi}(t + 1, x', u') \tag{37}$$

with terminal condition $Q_\alpha^{\boldsymbol{\mu},\pi}(T, x, u) \equiv 0$, which can be computed as in Algorithm 2, see also Puterman (2014) for a review.

To achieve convergence of fixed point iterations to approximate equilibria, for previous $\boldsymbol{\mu}^n \in \mathcal{M}$ we compute the action value function $Q_\alpha^{\boldsymbol{\mu}^n}$ via Algorithm 2 using approximate equivalence classes

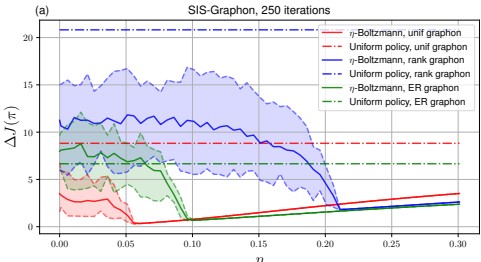 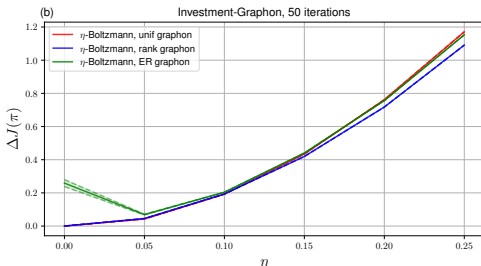

Figure 5: Final approximate exploitability mean and its minimum / maximum (shaded region) over the last 10 iterations for various temperatures $\eta$. We can see convergence for sufficiently high temperatures and choose the lowest temperature such that we still have convergence with low exploitability. Furthermore, compared to the uniformly random policy, our approximate exploitability is significantly lower, indicating a good approximate GMFE. For the Investment-Graphon problem, the approximate exploitability of the uniform policy is not shown, as it is above 30. **(a)**: SIS-Graphon; **(b)**: Investment-Graphon.

and then define the next policy $\boldsymbol{\pi}^{n+1} = \hat{\Phi}(\boldsymbol{\mu}^n)$ for every $\alpha \in \mathcal{I}$ via the softmax function

$$
\pi_t^{n+1,\alpha_i}(u \mid x) = \frac{\exp\left(\frac{Q_{\alpha_i}^{\boldsymbol{\mu}^n}(t,x,u)}{\eta}\right)}{\sum_{u \in \mathcal{U}} \exp\left(\frac{Q_{\alpha_i}^{\boldsymbol{\mu}^n}(t,x,u)}{\eta}\right)}
\tag{38}
$$

for the closest $\alpha_i$ with some temperature $\eta > 0$ chosen minimally for convergence.

For choosing the temperature, we evaluate the approximate final exploitability at various temperatures. The results can be seen in Figure 5, where we plot the average, minimum and maximum exploitability over the last 10 iterations of the fixed point learning scheme. The reasoning behind choosing our temperatures as in Table 2 is that we can see no fluctuations (indicating convergence of our learning scheme) together with a low approximate exploitability at the indicated temperatures.

### A.3.3 ADDITIONAL EXPERIMENTS

In Figure 6, we plot investment behavior at quality $x = 0$ as well as expected quality for each $\alpha$ of the approximate equivalence class solution, and similarly in Figure 7 for the PPO solution with sequential Monte Carlo. Here, for each $\alpha$ we averaged quality over all particles within a distance of 0.05 to $\alpha$. We can see that PPO achieves qualitatively and quantitatively similar behavior, deviating slightly due to the approximate optimality of the PPO algorithm. To be precise, when evaluating exploitability via either solution, we find that the learned policy exploitability remains around $\varepsilon \approx 2$, compared to $\varepsilon > 30$ for the uniform random policy.

In Figure 8 the equilibrium behavior is shown for the Investment-Graphon problem without softmax policy regularization (except for the ER graphon case), as we find that the problem already converges to a very good equilibrium with low approximate exploitability, see Figure 5. In this problem, we find that the resulting (deterministic without regularization) policy will let agents invest up to a certain quality, after which any further investment is avoided. The agents with higher connectivity will invest up to a lower quality, as they are in competition with more products.

In Figure 9 and 10, we have performed ablations over the number of equivalence classes for the SIS-Graphon problem. As can be observed, the solution obtained by approximate equivalence classes remains stable regardless of the particular number of equivalence classes, showing the stability of discretization approach and supporting Theorem 5.

Finally, in Figure 11 we exemplarily show training results of applying state-of-the-art multi-agent reinforcement learning methods such as multi-agent PPO (MAPPO, Yu et al. (2021)) on the finite-agent system with observed, randomized-per-episode graphon indices and $W$-random graphs. Here, we use the same hyperparameters as shown in Table 1. As can be seen, due to the non-stationarity of the other agents, a naive application of MARL techniques fails to converge at all.

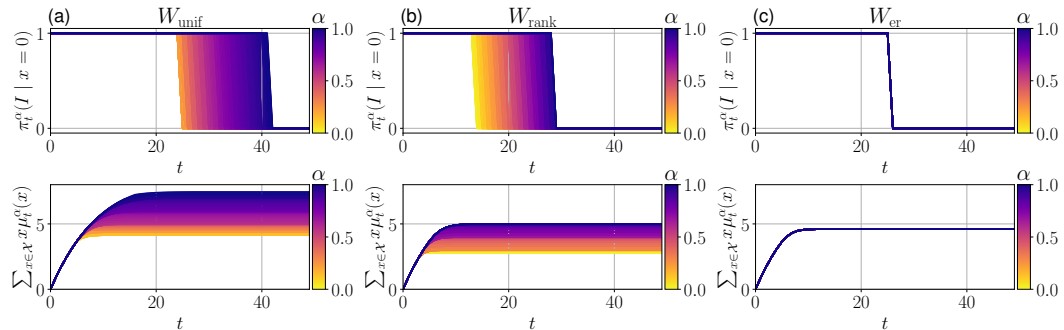

Figure 6: The $M = 100$ approximate equivalence classes solution of Investment-Graphon. We plot the probability of investing at state $x = 0$ (top) together with the evolution of average quality (bottom). **(a)**: Uniform attachment graphon; **(b)**: Ranked attachment graphon; **(c)**: ER graphon.

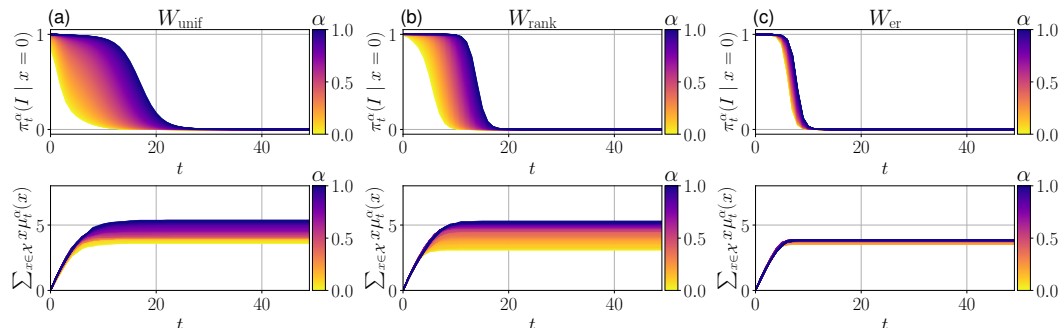

Figure 7: The probability of investing at state $x = 0$ (top) together with the evolution of average quality (bottom) for PPO. The solution is similar to Figure 6, though slightly different due to the approximations stemming from PPO and sequential Monte Carlo. **(a)**: Uniform attachment graphon; **(b)**: Ranked attachment graphon; **(c)**: ER graphon.

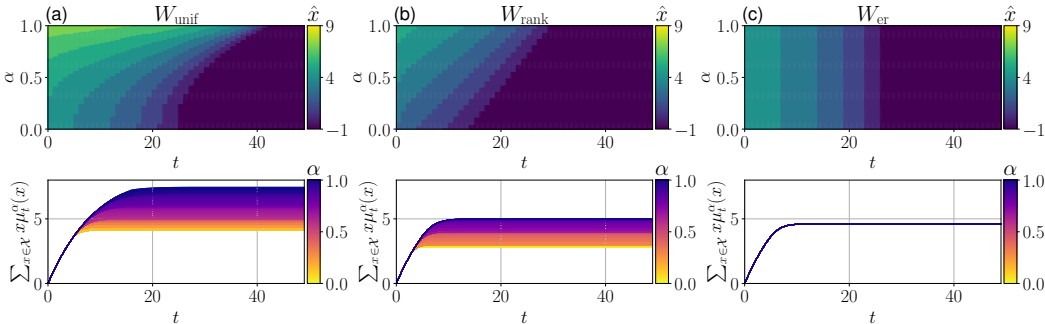

Figure 8: Achieved equilibrium via $M = 100$ approximate equivalence classes in Investment-Graphon. Top: Maximum quality $\hat{x}$ up to which agents will invest ($\pi_t^\alpha(I \mid \hat{x}) > 0.5$), shown for each $\alpha \in \mathcal{I}, t \in \mathcal{T}$. Bottom: Expected quality versus time of each agent $\alpha \in \mathcal{I}$. It can be observed that agents with less connections (higher $\alpha$) will invest more. **(a)**: Uniform attachment graphon; **(b)**: Ranked attachment graphon; **(c)**: ER graphon.

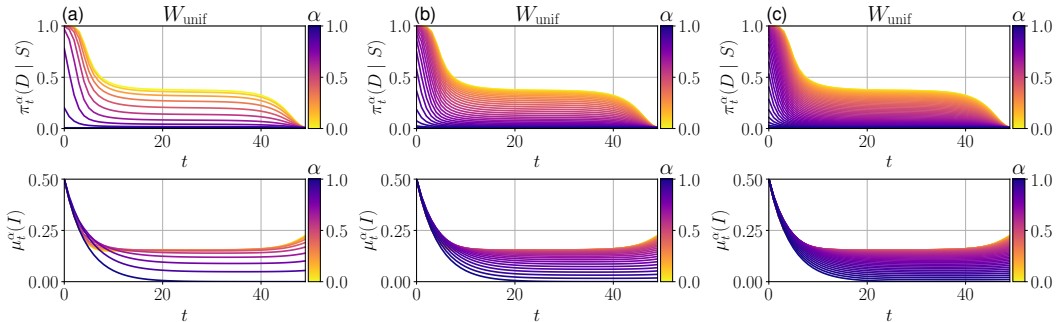

Figure 9: Achieved equilibrium via approximate equivalence classes in SIS-Graphon for the uniform attachment graphon, plotted for each representative $\alpha_i \in \mathcal{I}$. Top: Probability of taking precautions when healthy. Bottom: Probability of being infected. **(a)**: $M = 10$; **(b)**: $M = 30$; **(c)**: $M = 50$.

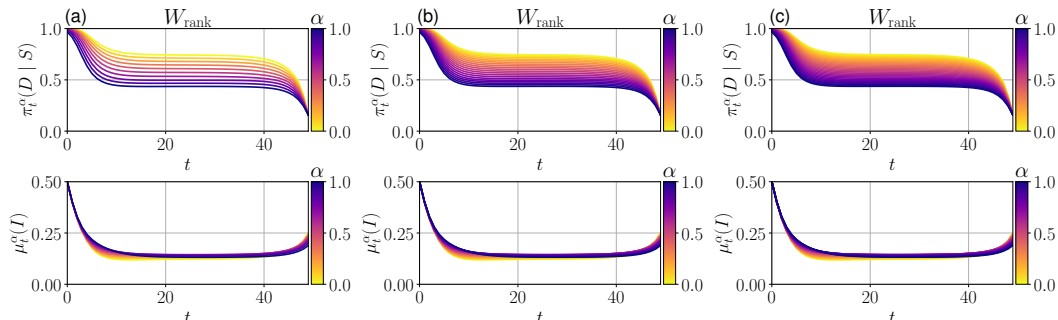

Figure 10: Achieved equilibrium via approximate equivalence classes in SIS-Graphon for the ranked attachment graphon, plotted for each representative $\alpha_i \in \mathcal{I}$. Top: Probability of taking precautions when healthy. Bottom: Probability of being infected. **(a)**: $M = 10$; **(b)**: $M = 20$; **(c)**: $M = 30$.

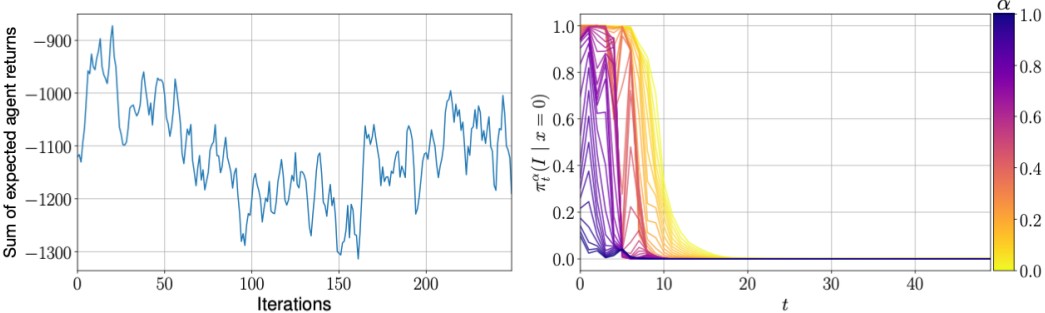

Figure 11: Learning curve and results for an exemplary straightforward application of multi-agent PPO (MAPPO, Yu et al. (2021)). Left: Sum of expected agent objectives over learning iterations; Right: Final policy probability of taking precautions when healthy.

