# OpenReview forum: "Learning Graphon Mean Field Games and Approximate Nash Equilibria"
_ICLR.cc/2022/Conference — ICLR 2022 Poster_

### Official Review · Reviewer_B2QF · 2021-11-01

**Correctness:** 4
**Technical Novelty And Significance:** 3
**Empirical Novelty And Significance:** 4
**Recommendation:** 8
**Confidence:** 3

**Main Review:**

Overall, I think the paper makes a strong and important contribution. To the best of my knowledge, the results in this work are novel: the formulation for graphon mean field games is new, the existence and convergence results are new, and the algorithm is novel as well. The problem considered in the paper is an important problem in practice: in many applications, the subjects of interest interact with each other. The authors provide a strong solution to the problem, from modeling, to theory, to algorithm. The paper is also well-written and I enjoyed reading it.

I don’t see a clear weakness of the paper. Here are a few things I’d love to see (which of course might be beyond the scope of this work due to page limit)
-	Sparse graphon, i.e., the graphon W_N / rho_N -> W, where rho_N is the sparsity parameter that goes to 0 as N goes to infinity. As far as I checked, I think the proofs can still go through in this case (as long as rho_N is not too small). I think this might be a more realistic setting in practice, since we won’t be interacting with a constant proportion of people in the world.
-	Is it possible to obtain a result on convergence rate for Proposition 2?


**Summary Of The Paper:**

This paper studies Markov games with large number of agents. The authors start by proposing a novel discrete-time formulation for graphon mean field games. They show that the mean field game exists, and that finite graph games converge to the mean field game when the number of agents goes to infinity. They also provide an algorithm that learns an approximate Nash equilibria.

**Summary Of The Review:**

Overall, I think the paper makes a strong and important contribution. The results in this work are novel and important. The paper is well-written.

---

> ### Author Response · Authors · 2021-11-17
> **Official Response to Review of Paper1361 by Reviewer B2QF (1/1)**
>
> We thank the reviewer for their detailed reading as well as positive feedback. We believe that the points raised by all reviewers have been addressed, in turn significantly strengthening our paper as well as our algorithmic contribution. Please note that the length of the paper has increased by 6 pages in order to address all feedback.
>
> ## Regarding the raised points
> > - Sparse graphon, i.e., the graphon W_N / rho_N -> W, where rho_N is the sparsity parameter that goes to 0 as N goes to infinity. As far as I checked, I think the proofs can still go through in this case (as long as rho_N is not too small). I think this might be a more realistic setting in practice, since we won’t be interacting with a constant proportion of people in the world.
>
> Indeed, the case with additional sparsity parameters such as $\beta_N$ in [A, Theorem 4.2] may be considered for further generalization of obtained results. As the focus of our paper lies on solving graphon mean field games and such a generalization is associated with significant additional analysis, we defer an according extension of analysis to future work.
>
>
> > - Is it possible to obtain a result on convergence rate for Proposition 2?
>
> In general, for contractions as given in Proposition 2, we are automatically guaranteed an at least linear convergence rate by definition of a contraction. Whether improved convergence rates are possible currently remains outside the scope of this work and is an interesting future direction.
>
> ## Other
>
> In response to the feedback received by all reviewers, we have updated our paper and posted a list of changes in a separate comment. To summarize, most notably we have added
>
> 1. a formal convergence theory for the equivalence classes method,
> 2. a formal proof of optimality for our equivalence classes method,
> 3. further experiments ablating over the discretization grids and comparing to MARL,
> 4. added helpful introductory visualizations of the considered graphical dynamics, and
> 5. discussions in response to each of the points raised by the reviewers.
>
> [A] Bayraktar, Erhan, Suman Chakraborty, and Ruoyu Wu. "Graphon mean field systems." arXiv preprint arXiv:2003.13180 (2020).

---

### Official Review · Reviewer_V2M1 · 2021-11-02

**Correctness:** 4
**Technical Novelty And Significance:** 3
**Empirical Novelty And Significance:** 2
**Recommendation:** 5
**Confidence:** 4

**Main Review:**

1. Proposition 2 is very standard in the MFG literature. However, the assumption seems very abstract and too strong. The authors should give some discussions on when the assumption holds or how to verify this assumption, especially the existence of the Lipschitz mapping $\hat{\Phi}$, which is an element chosen from a set mapping (e.g., how it is chosen). Some more formal and detailed explanations are needed here.
2. The first algorithm for learning GMFG in Section 4 is not well-explained. More details should be provided. In the first method in Section 4, it’s not clear why it makes sense to discretize $\alpha$ into $\alpha_i$’s and solve optimal control problems for each equivalence class of the associated $\alpha_i$. More formal justifications of this algorithm is needed to understand how it’s derived, especially given that it has no convergence analysis.
3. The algorithms used in the experiments can not solve exactly for the NE of the GMFG problem since they all use discretization. And the error induced by this discretization is not mentioned in the paper. It is also not clear what discretization (the value of $M$) is chosen in Algorithm 2 and 3. If we want to find the NE for GMFG, it would be better if experiments can show how solution evolves when $M$ gets larger instead of fixing one value of $M$.
4. There is no theoretical convergence or related performance analysis regarding the two proposed algorithm in Section 4, which largely limited the paper's algorithmic contributions. Proposition 2 is mostly an existing result in the classical learning MFG literature, and neglects all kinds of randomness and approximation, which is not very meaningful.
5. The numerical results lack some comparisons with existing MARL algorithms (with and without mean-field approximations, e.g., those in Yang et al., 2018).

**Summary Of The Paper:**

This paper studies discrete time dense graph mean field games (GMFG). It discusses the settings of both finite-agent graph game and GMFG and shows that under mild Lipschitz continuous assumptions, the Nash equilibrium of the GMFG exists. It establishes that the NE of the GMFG is a good approximation of the NE of an N-agent game and empirically verifies this result. It then proposes two algorithms for solving the NE of GMFG, one with the idea of discretizing the graphon index, and the other utilizing the equivalence of GMFG to a classical MFG with an extended state space. The proposed algorithms are tested on an SIS-Graphon problem and an Investment-Graphon problem to validate their performances.

**Summary Of The Review:**

This paper provides some nice contributions to the theory of graphon mean-field games in discrete-time settings. It also provides some heuristic learning algorithms, though without theoretical justifications. Some preliminary numerical results are also provided, but without comparisons with the existing MARL algorithms.

---

> ### Author Response · Authors · 2021-11-17
> **Official Response to Review of Paper1361 by Reviewer V2M1 (2/2)**
>
> ...
>
> > 4. There is no theoretical convergence or related performance analysis regarding the two proposed algorithm in Section 4, which largely limited the paper's algorithmic contributions. Proposition 2 is mostly an existing result in the classical learning MFG literature, and neglects all kinds of randomness and approximation, which is not very meaningful.
>
> We thank the reviewer for the constructive suggestion and have added a formal convergence theory for our algorithm. We have added the corresponding result in Section 4 as Theorem 4, as well as a rigorous error analysis as Theorem 5.
>
> Note that the approach via equivalence classes is not stochastic, unless reinforcement learning is used to solve the control problem. For the approximate reinforcement learning method, an analysis would go beyond the scope of our work, as significant further analysis would be required.
>
> > 5. The numerical results lack some comparisons with existing MARL algorithms (with and without mean-field approximations, e.g., those in Yang et al., 2018).
>
> We agree partially, though we would like to point out that our work focuses on using RL to solve the limiting system, and does not yet constitute a MARL method to be applied to arbitrary finite-agent systems. The important difference here is that the limiting mean field problem only models a representative agent (with graphon index as state) together with a mean field term, i.e. we apply single-agent reinforcement learning to solve the limiting system, more similar to works such as [B, C, D] rather than [E]. For MARL, the additional step of graphon estimation is an important step, as discussed in the conclusion and Section 4.
>
> We have further added a formal proof of optimality for our equivalence classes method (Theorem 5). Together with the added ablation over the number $M$ of equivalence classes, which shows stability of our proposed solution over $M$, we believe our analysis to suffice, as there would be little point in comparing approximate MARL methods with an optimal solution. Nonetheless, we have added an exemplary experiment using MARL methods at the end of the appendix, where due to the non-stationarity of the other agents, a naive application of state-of-the-art MARL techniques fails to converge, while the mean field MARL approach in (Yang et al., 2018) remains incomparable due to necessity of observing the average previous numerical actions of other agents, whereas our setting remains fully decentralized (local state only).
>
> ## Other
>
> In response to the feedback received by all reviewers, we have updated our paper and posted a list of changes in a separate comment. To summarize, most notably we have added
>
> 1. a formal convergence theory for the equivalence classes method,
> 2. a formal proof of optimality for our equivalence classes method,
> 3. further experiments ablating over the discretization grids and comparing to MARL,
> 4. added helpful introductory visualizations of the considered graphical dynamics, and
> 5. discussions in response to each of the points raised by the reviewers.
>
> [A] Cui, Kai, and Heinz Koeppl. "Approximately solving mean field games via entropy-regularized deep reinforcement learning." International Conference on Artificial Intelligence and Statistics. PMLR, 2021.
>
> [B] Guo, Xin, et al. "Learning Mean-Field Games." Advances in Neural Information Processing Systems 32 (2019): 4966-4976.
>
> [C] Elie, R., Perolat, J., Laurière, M., Geist, M., \& Pietquin, O. (2020, April). On the convergence of model free learning in mean field games. In Proceedings of the AAAI Conference on Artificial Intelligence (Vol. 34, No. 05, pp. 7143-7150).
>
> [D] Subramanian, Jayakumar, and Aditya Mahajan. "Reinforcement learning in stationary mean-field games." Proceedings of the 18th International Conference on Autonomous Agents and MultiAgent Systems. 2019.
>
> [E] Yang, Yaodong, et al. "Mean field multi-agent reinforcement learning." International Conference on Machine Learning. PMLR, 2018.

---

> ### Author Response · Authors · 2021-11-17
> **Official Response to Review of Paper1361 by Reviewer V2M1 (1/2)**
>
> We thank the reviewer for their detailed reading as well as highly constructive comments. We believe that the points raised by the reviewer have been addressed, in turn significantly strengthening our paper as well as our algorithmic contribution. Please note that the length of the paper has increased by 6 pages in order to address all feedback.
>
> ## Regarding the major concerns
>
> > 1. Proposition 2 is very standard in the MFG literature. However, the assumption seems very abstract and too strong. The authors should give some discussions on when the assumption holds or how to verify this assumption, especially the existence of the Lipschitz mapping $\hat \Phi$, which is an element chosen from a set mapping (e.g., how it is chosen). Some more formal and detailed explanations are needed here.
>
> Proposition 2 is indeed standard in the literature. However, we have chosen $\hat \Phi$ as indicated in Section 4, i.e. $\hat \Phi$ is not the set-valued map $\Phi$ and instead is realized either through reinforcement learning, or through the choice of (near-)optimal policies for a representative of each equivalence class. In particular, note that a softmax policy choice of $\hat \Phi$ is Lipschitz continuous, while the map from policy to mean field $\hat \Psi$ is automatically Lipschitz, and therefore we are indeed able to obtain a formal convergence theory for our algorithm. We have improved the description of our algorithms and added the corresponding discussion in Section 4 as Theorem 4.
>
> > 2. The first algorithm for learning GMFG in Section 4 is not well-explained. More details should be provided. In the first method in Section 4, it’s not clear why it makes sense to discretize $\alpha$ into $\alpha_i$’s and solve optimal control problems for each equivalence class of the associated $\alpha_i$. More formal justifications of this algorithm is needed to understand how it’s derived, especially given that it has no convergence analysis.
>
> Regarding the description of our algorithms, we have significantly improved upon the description and added a significant amount of details. As a result, Section 4 has been expanded significantly, and some experiments were moved to the Appendix for space reasons. In particular, we have added a rigorous justification for the algorithm design in the form of Theorems 4 and 5, showing that the discretized class of solutions can achieve exactly the same type of result as an exact solution.
>
> > 3. The algorithms used in the experiments can not solve exactly for the NE of the GMFG problem since they all use discretization. And the error induced by this discretization is not mentioned in the paper. It is also not clear what discretization (the value of $M$) is chosen in Algorithm 2 and 3. If we want to find the NE for GMFG, it would be better if experiments can show how solution evolves when $M$ gets larger instead of fixing one value of $M$.
>
> We would like to point out that the second, direct reinforcement learning method we proposed does not use discretization and in principle can solve the GMFG with arbitrary precision. We furthermore thank the reviewer for the constructive suggestion and have added an ablation study on the number of discretization intervals (the value $M$) in Appendix A.3.3 to discuss the effect of the discretization setting. As can be seen from the experiment, the approach remains empirically stable over a wide range of discretization settings, showing the stability of our discretization approach. Together with an added formal proof of optimality for our equivalence classes method in Section 4, as well as a formal convergence theory for our algorithm in Theorem 4, we believe to have improved our analysis of the algorithm based on the feedback. In particular, the discretized solution comes arbitrarily close to optimality, as shown by the added Theorem 5.
>
> As a side remark, although we use approximations to solve the limiting GMFG, even an exact solution of the GMFG only constitutes an approximate Nash equilibrium in the finite-graph system, and this holds similarly when solving classical MFGs, see e.g. [A]. Therefore, we argue that little is lost by introducing additional small approximations for the sake of a tractable solution. We have added the corresponding discussion to the end of Section 4.
>
> ...

---

### Official Review · Reviewer_wpJP · 2021-11-02

**Correctness:** 4
**Technical Novelty And Significance:** 3
**Empirical Novelty And Significance:** 2
**Recommendation:** 6
**Confidence:** 4

**Main Review:**

Graphon games are an emerging topic of research and developing learning algorithms for such games is an interesting research direction. Furthermore, the authors provide rigorous proofs for the statements of their main results. The proof techniques are quite classical in the mean field game literature, but it is nice that the authors provide detailed proofs in the appendix.

My concerns are the following:

1. Definition 1: It would be helpful for the reader to clarify why you use this notion, which is weaker than the simpler and more classical notion of $\epsilon$-Nash equilibrium. Is there a classical reference for this notion? By the way, is it connected to the notion of $(\epsilon,\delta)$ mean field Nash equilibrium considered (for the limiting game) e.g. in (Elie et al., 2020)?

2. Theorems 2 and 3: the Lipschitz condition on $\boldsymbol{\pi}$ seems very restrictive. The authors state that it is easy to verify it in “the case where only finitely many optimality regimes exist over all graphon indices”. Although this property probably holds for block-wise constant graphons, I imagine it is not satisfied in many interesting examples. It would be helpful to formulate conditions on the model (transition and reward functions) that are sufficient to obtain this Lipschitz continuity.

3. About the computational advantage of GMFG: below Theorem 3, the authors recall that computing Nash equilibria in finite-player games is in general "highly intractable". However, here the authors consider a special class of games. Could you quantify the computational advantage brought by the graphon game framework compared with its N-player counterpart? It seems that (1) the class of graphon games studied in this paper is quite restrictive due to the regularity assumptions and the fact that the transition and reward functions are the same functions for all players and (2) the authors use extra approximations in the two proposed numerical approaches so we can imagine that these methods compute only approximations of Nash equilibria. So overall, I am not sure whether the graphon games considered in this work are really more tractable than their N-player counterparts. It would be very important to explain this better in order to clarify the advantages of the proposed approaches.

4. About the numerical examples: For the first example (SIS-example in section 5.1), the results are interesting but there is not clear description of which algorithm is being used. In Figure 7 (appendix), it is nice to see the influence of the temperature. But sometimes the temperature required to get convergence is not negligible. I imagine that due to this issue, the solution can be quite different from the true Nash equilibrium. So do we know the influence of this temperature on the policy computed by the algorithm?

Reference:
(Elie et al., 2020) Elie, R., Perolat, J., Laurière, M., Geist, M., & Pietquin, O. (2020, April). On the convergence of model free learning in mean field games. In Proceedings of the AAAI Conference on Artificial Intelligence (Vol. 34, No. 05, pp. 7143-7150).

**Summary Of The Paper:**

This paper studies a class of games with a continuum of agents that connected by a graphon. This corresponds to the limit of games with a finite number of players connected by a graph. In the recent literature, such games have been considered in continuous time, and here the authors focus on a discrete time version. They first analyze the game by showing existence of a Nash equilibrium (defined in a suitable sense), and they prove that using the equilibrium policy from the game with a continuum of players in a game with a finite number of players provides an approximate Nash equilibrium. Then, two numerical approaches are proposed, both based on fixed point iterations that alternate between updating the distribution and updating the optimal control. The first method relies on backward induction to compute the optimal control of representative agents in a set of equivalence classes. The second method relies on reinforcement learning to compute the optimal control. Last, numerical examples are provided.

**Summary Of The Review:**

The authors provide theoretical results showing the game is well posed, and they provide interesting numerical results (also in the appendix). But overall I am not sure whether the “learning” part is really crucial in the paper. If it is the case, more explanations on the advantages of the proposed approach should be included.

---

> ### Author Response · Authors · 2021-11-17
> **Official Response to Review of Paper1361 by Reviewer wpJP (3/3)**
>
> ...
>
> ## Other
>
> Regarding the importance and meaning of learning in our work: In our work, learning does not only refer to reinforcement learning, but also to the classical notion of learning Nash equilibria which is not inherently sample-based, see e.g. [E]. We would also like to point out that our work focuses on using RL to solve a limiting system, and not only MARL for multi-agent systems. Further, while it is possible to exactly solve optimal control problems for each agent equivalence class with finite state-action spaces, this is generally not the case for e.g. continuous state-action spaces. Here, a reinforcement-learning-based solution can solve otherwise intractable problems in an elegant manner, since the graphon index $\alpha$ simply becomes part of a continuous state space. This allows for great generality and elegance of our proposed approach, since it is indeed also possible to incorporate e.g. heterogeneous types of agents into the state. We have added the corresponding discussion to Section 4.
>
> In response to the feedback received by all reviewers, we have updated our paper and posted a list of changes in a separate comment. To summarize, most notably we have added
>
> 1. a formal convergence theory for the equivalence classes method,
> 2. a formal proof of optimality for our equivalence classes method,
> 3. further experiments ablating over the discretization grids and comparing to MARL,
> 4. added helpful introductory visualizations of the considered graphical dynamics, and
> 5. discussions in response to each of the points raised by the reviewers.
>
> [A] Saldi, Naci, Tamer Basar, and Maxim Raginsky. "Markov--Nash Equilibria in Mean-Field Games with Discounted Cost." SIAM Journal on Control and Optimization 56.6 (2018): 4256-4287.
>
> [B] Mondal, Washim Uddin, et al. "On the Approximation of Cooperative Heterogeneous Multi-Agent Reinforcement Learning (MARL) using Mean Field Control (MFC)." arXiv preprint arXiv:2109.04024 (2021).
>
> [C] Pasztor, Barna, Ilija Bogunovic, and Andreas Krause. "Efficient Model-Based Multi-Agent Mean-Field Reinforcement Learning." arXiv preprint arXiv:2107.04050 (2021).
>
> [D] Cui, Kai, and Heinz Koeppl. "Approximately solving mean field games via entropy-regularized deep reinforcement learning." International Conference on Artificial Intelligence and Statistics. PMLR, 2021.
>
> [E] Daskalakis, Constantinos, et al. "On learning algorithms for Nash equilibria." International Symposium on Algorithmic Game Theory. Springer, Berlin, Heidelberg, 2010.
>
> [F] Carmona, Guilherme. Nash Equilibria of Games with a Continuum of Players. No. 0412009. University Library of Munich, Germany, 2004.
>
> [G] Bayraktar, Erhan, Suman Chakraborty, and Ruoyu Wu. "Graphon mean field systems." arXiv preprint arXiv:2003.13180 (2020).

---

> ### Author Response · Authors · 2021-11-17
> **Official Response to Review of Paper1361 by Reviewer wpJP (2/3)**
>
> ...
>
> > 3. About the computational advantage of GMFG: below Theorem 3, the authors recall that computing Nash equilibria in finite-player games is in general "highly intractable". However, here the authors consider a special class of games. Could you quantify the computational advantage brought by the graphon game framework compared with its N-player counterpart? It seems that (1) the class of graphon games studied in this paper is quite restrictive due to the regularity assumptions and the fact that the transition and reward functions are the same functions for all players and (2) the authors use extra approximations in the two proposed numerical approaches so we can imagine that these methods compute only approximations of Nash equilibria. So overall, I am not sure whether the graphon games considered in this work are really more tractable than their N-player counterparts. It would be very important to explain this better in order to clarify the advantages of the proposed approaches.
>
> For quantification of the complexity, we would like to point out that no dynamic programming principle will hold in finite-$N$ games due to the non-Markovian local agent state and decentralized nature of our solution (local feedback policies without memory), see also the discussion in [A, page 5] and Section 5, while alternatively acting on the full state fails by the curse of dimensionality, as the state blows up exponentially in the number of agents $N$. Meanwhile, the mean field formulation remains independent of $N$ and thus its complexity remains incomparable to the exponential-in-$N$ complexity of the finite-$N$ case.
>
> Regarding (1), the (Lipschitz) continuity assumption is standard in MFG literature and strictly required for weak interaction of agents, see e.g. [A, G]. Further, note that the heterogeneous agents case is included by modelling agent types as part of the agent state. It is only required to model agent states in a unified manner, which does not imply that there cannot be heterogeneity among agents. We have added the corresponding discussion to Section 2 as Remark 2. We can also consider infinite-horizon formulations, time-dependent functions and heterogeneous starting conditions, as the proofs will hold and are written with this in mind, see also Remark 1.
>
> Regarding (2), although we use approximations to solve the limiting GMFG, even an exact solution of the GMFG only constitutes an approximate Nash equilibrium in the finite-graph system, and this holds similarly when solving classical MFGs, see e.g. [A]. Therefore, we argue that little is lost by introducing additional small approximations for the sake of a tractable solution, in particular since we have also added a formal proof of optimality for our equivalence classes method to justify the arbitrary exactness of the discretization approach. We have added the corresponding discussion to the end of Section 4.
>
> > 4. About the numerical examples: For the first example (SIS-example in section 5.1), the results are interesting but there is not clear description of which algorithm is being used. In Figure 7 (appendix), it is nice to see the influence of the temperature. But sometimes the temperature required to get convergence is not negligible. I imagine that due to this issue, the solution can be quite different from the true Nash equilibrium. So do we know the influence of this temperature on the policy computed by the algorithm?
>
> The caption of Figure 3 mentions approximate equivalence classes, but we have clarified this by adding information on the number of equivalence classes. The suboptimality (exploitability) stemming from the temperature is given exactly by the $\Delta J$ in Figure 5 (of the new PDF). Note that as seen in the figure, the achievable exploitability is quite low as compared to suboptimal solutions such as the uniform policy. Regarding the theoretical influence of the temperature, it has been shown in [D] for classical MFGs that the limiting exploitability $\epsilon$ tends to zero with the temperature, i.e. the influence of the temperature on the policy and its suboptimality is at least continuous. A more precise analysis of the influence of temperature on suboptimality remains to the best of our knowledge an open future research direction.
>
> ...

---

> ### Author Response · Authors · 2021-11-17
> **Official Response to Review of Paper1361 by Reviewer wpJP (1/3)**
>
> We thank the reviewer for their detailed reading as well as highly constructive comments. We believe that the points raised by the reviewer have been addressed, in turn significantly strengthening our paper as well as our algorithmic contribution. Please note that the length of the paper has increased by 6 pages in order to address all feedback.
>
> ## Regarding the major concerns
>
> > 1. Definition 1: It would be helpful for the reader to clarify why you use this notion, which is weaker than the simpler and more classical notion of -Nash equilibrium. Is there a classical reference for this notion? By the way, is it connected to the notion of $(\epsilon, \delta)$ mean field Nash equilibrium considered (for the limiting game) e.g. in (Elie et al., 2020)?
>
> We thank the reviewer for pointing out the related work (Elie et al., 2020) and have added it to the related works section. Indeed, it appears that our considered notion of $(\epsilon, p)$-Nash equilibria coincides with the $(\epsilon, \delta)$-Nash equilibria in (Elie et al., 2020) and [F], and we have added the corresponding references to the paper. We have also further discussed the reason of introducing the reason for this notion of Nash equilibria: Under graph convergence in Assumption 1, it is always possible for a finite number of nodes to have an arbitrary neighborhood differing from the graphon as $N \to \infty$. Thus, it is impossible to show approximate optimality for all nodes and only possible to show for an increasingly large fraction $1-p \to 1$ of nodes. For this reason, we slightly weaken the notion of Nash equilibria by restricting to a fraction $1-p$ of agents, which is sufficient for optimality of all other agents, as by weak interaction all other nodes in fraction $p \to 0$ become irrelevant.
>
> > 2. Theorems 2 and 3: the Lipschitz condition on $\boldsymbol \pi$ seems very restrictive. The authors state that it is easy to verify it in “the case where only finitely many optimality regimes exist over all graphon indices”. Although this property probably holds for block-wise constant graphons, I imagine it is not satisfied in many interesting examples. It would be helpful to formulate conditions on the model (transition and reward functions) that are sufficient to obtain this Lipschitz continuity.
>
> Regarding the restrictivity of the assumption of block-wise Lipschitz policies, at least for our numerical methods using finitely many equivalence classes or typical neural network policies, the assumption holds, similar to the recent publications [B, Assumption 3] and [C, Assumption 2]. Most importantly, the condition holds for the equivalence classes method, for which we have added a formal proof of optimality for our equivalence classes method, which implies that block-wise Lipschitz policies are indeed sufficient for achieving the same results as an exact solution. As shown formally in Theorem 5, there always exist such Lipschitz policies that are arbitrarily close to a Nash equilibrium, i.e. the block-wise Lipschitz policies we consider are indeed sufficient for our purposes of learning an $\epsilon$-Nash equilibrium. Whether there exist non-block-wise Lipschitz policies that also constitute an $\epsilon$-Nash equilibrium is an interesting but separate question. We have added the corresponding discussion and references to the paper.
>
> ...

---

### Official Review · Reviewer_GRFn · 2021-11-07

**Correctness:** 3
**Technical Novelty And Significance:** 3
**Empirical Novelty And Significance:** 2
**Recommendation:** 5
**Confidence:** 4

**Main Review:**

[Major Pros]

This paper proposes a novel GMFG framework, with a relatively complete characterization of the GMFE existence and graphon mean-field approximation errors. The proposed learning algorithms and the numerical experiments also add to the practical contribution of this paper.

[Major Cons]

Despite the several pros mentioned above, there are also some issues that the authors need to address and/or improve on, as listed below.
1. Section 2 is a bit too heavy as an introductory section. It might be better to start with Section 2.1, and describe the idea of local interaction and local averaging for a group of networked agents in a more intuitive manner. Some visualization of the N-agent scenario will also be helpful (see e.g., (Yang et al., 2018)).
2. The algorithm description and motivation of the first learning method (which discretizes the graphon index) are not sufficiently clear. It would be helpful to compare this discretization approach with multi-population mean-field games in more details and discuss about what are the benefits of considering GMFG, and to better explain why for each equivalence class the optimal control problems can be solved independently. The verbal description is a bit too informal, while the Appendix A.3 is too focused on the experiment details while not providing sufficient explanations on how these heuristic algorithms are derived. The justification of the algorithm design logic is especially important since no formal convergence theory for the proposed learning algorithms is established.
3. The numerical experiments lack some important comparisons. Firstly, since GMFG is proposed to solve N-player games approximately, existing algorithms for solving N-player games (especially those proposed for multi-agent reinforcement learning) are all fair comparisons. Hence to demonstrate the effectiveness of the proposed two learning schemes, comparisons with algorithms such as those mentioned in (Yang et al., 2018) and [A] are needed. Also, performance in the N-player setting is only demonstrated with the indirect metric (26), which is not sufficiently convincing and informative.

There are also some more minor comments and questions, as detailed below:
1. The authors should compare with [B].
2. On page 2, should “Graph mean field systems” be “Graphon mean field systems”?
3. Please discuss more about the relationship between the GMFG model and the one adopted in (Yang et al., 2018) and [C], which are also related to graph based games.
4. In Section 5.2, do you mean that (26) instead of (19) is evaluated? And what is \pi^{\alpha_i} and how is it computed?
5. In Section 6, what does “graphon index estimation become important for finite system model-based RL” mean?
6. In Appendix A.3.1, what is G(I)?

[A] Lowe, Ryan, Yi Wu, Aviv Tamar, Jean Harb, Pieter Abbeel, and Igor Mordatch. "Multi-agent actor-critic for mixed cooperative-competitive environments." arXiv preprint arXiv:1706.02275 (2017).

[B] Vasal, Deepanshu, Rajesh Mishra, and Sriram Vishwanath. "Sequential decomposition of graphon mean field games." In 2021 American Control Conference (ACC), pp. 730-736. IEEE, 2021.

[C] Yang, Jiachen, Xiaojing Ye, Rakshit Trivedi, Huan Xu, and Hongyuan Zha. "Learning deep mean field games for modeling large population behavior." arXiv preprint arXiv:1711.03156 (2017).


**Summary Of The Paper:**

This paper proposes a discrete-time graphon mean-field game (GMFG) framework as a limiting model to approximate large population dense graph games. Existence of graphon mean-field equilibrium (GMFE) is proved by reformulating GMFG as a classical MFG, and a graphon mean-field approximation result is also obtained. The authors then propose two learning algorithms for computing GMFE, one based on discretizing the graphon index, while the other based on the aforementioned classical MFG reformulation. Some preliminary numerical experiments are also provided to demonstrate the theoretical findings and to validate the proposed algorithms.

**Summary Of The Review:**

The contribution on proposing discrete-time GMFG and establishing existence and approximation results are good, but the algorithmic and numerical contributions are limited.

---

> ### Author Response · Authors · 2021-11-17
> **Official Response to Review of Paper1361 by Reviewer GRFn (3/3)**
>
> ...
>
> > 4. In Section 5.2, do you mean that (26) instead of (19) is evaluated? And what is \pi^{\alpha_i} and how is it computed?
>
> We have evaluated (26) as an upper bound on (19) by using the GMFE policy, i.e. we show that the finite system is well approximated by the mean field system. See also the response to major concern 3. With $\pi^{\alpha_i}$ we denote the GMFE policy in Appendix A.3 computed by the equivalence class method in Section 4 for the $\alpha_i$ closest to the deviating agent. To make it more clear, we have changed the formulation in the main text accordingly.
>
> > 5. In Section 6, what does “graphon index estimation become important for finite system model-based RL” mean?
>
> We meant that the estimation of the limiting graphon and agent graphon indices for a given finite-graph system is important for a direct application of our methodology in finite-graph systems, i.e. as a direct MARL method. We have reformulated the sentence accordingly.
>
> > 6. In Appendix A.3.1, what is G(I)?
>
> We thank the reviewer for pointing out a missing definition. In Appendix A.3.1, $\mathbb G$ denotes a fixed neighborhood mean field, i.e. $\mathbb G(I)$ denotes the (scaled) total number of infected (I) neighbors. We have added the definition at the beginning of Appendix A.3.1.
>
> ## Other
>
> In response to the feedback received by all reviewers, we have updated our paper and posted a list of changes in a separate comment. To summarize, most notably we have added
>
> 1. a formal convergence theory for the equivalence classes method,
> 2. a formal proof of optimality for our equivalence classes method,
> 3. further experiments ablating over the discretization grids and comparing to MARL,
> 4. added helpful introductory visualizations of the considered graphical dynamics, and
> 5. discussions in response to each of the points raised by the reviewers.
>
> [A] Subramanian, Jayakumar, and Aditya Mahajan. "Reinforcement learning in stationary mean-field games." Proceedings of the 18th International Conference on Autonomous Agents and MultiAgent Systems. 2019.
>
> [B] Vasal, Deepanshu, Rajesh Mishra, and Sriram Vishwanath. "Sequential decomposition of graphon mean field games." In 2021 American Control Conference (ACC), pp. 730-736. IEEE, 2021.
>
> [C] Yang, Jiachen, Xiaojing Ye, Rakshit Trivedi, Huan Xu, and Hongyuan Zha. "Learning deep mean field games for modeling large population behavior." arXiv preprint arXiv:1711.03156 (2017).
>
> [D] Elie, R., Perolat, J., Laurière, M., Geist, M., \& Pietquin, O. (2020, April). On the convergence of model free learning in mean field games. In Proceedings of the AAAI Conference on Artificial Intelligence (Vol. 34, No. 05, pp. 7143-7150).
>
> [E] Guo, Xin, et al. "Learning Mean-Field Games." Advances in Neural Information Processing Systems 32 (2019): 4966-4976.

---

> > ### Comment · Reviewer_GRFn · 2021-12-06
> > **Thanks for the response but found additional issues in the revision**
> >
> > I sincerely appreciate the authors for the efforts in improving the paper, but unfortunately, I actually found some additional issues in the revision.
> >
> > In particular, although the authors added convergence results for the equivalence class algorithm, there are some obvious issues in the statements. For example, in the updated Proposition 2, the results imply that the fixed-point of the composition of $\hat{\Phi}$ and $\hat{\Psi}$ is a GMFE, which is not true in general for the approximations $\hat{\Phi}$ and $\hat{\Psi}$ of $\Phi$ and $\Psi$, respectively. Also, in Theorem 4, the notation $\hat{\Phi}(\mu)_t^{\alpha}(u|x)$ is weird, and in Theorem 5, $\hat{\Phi}$ is not clearly defined. I guess the authors probably meant to use (38) in the appendix for defining $\pi^{\alpha_i}$ in the $\hat{\Phi}$ of the equivalence class algorithm, but a clear and complete algorithm description for the equivalence class algorithm and fixing the statements in the new proposition and theorems are needed.
> >
> > Moreover, the new comparison with MAPPO is not convincing. In fact, MAPPO is for cooperative MARL, while the method in the current paper is for competitive MARL. Hence I don't see why it's fair to compare MAPPO here. Moreover, the authors' explanations (such as decentralization, etc.) about why they don't compare with other (competitive) MARL algorithms are not very convincing. In fact, many popular competitive MARL algorithms (including those compared in (Yang et al., 2018)) are indeed fully decentralized in the sense that each agent only reacts to its own state, including MADDPG [A] and [B], to name just a few.
> >
> > Hence after rethinking the pros (existence and approximation results) and cons (algorithmic and numerical contributions) more carefully, I decide to slightly lower my score to 5. Overall, I still think this is a promising work but does need some substantial polishing before publishing.
> >
> > [A] Lowe, Ryan, Yi Wu, Aviv Tamar, Jean Harb, Pieter Abbeel, and Igor Mordatch. "Multi-agent actor-critic for mixed cooperative-competitive environments." arXiv preprint arXiv:1706.02275 (2017).
> >
> > [B] Qu, Guannan, Adam Wierman, and Na Li. "Scalable reinforcement learning of localized policies for multi-agent networked systems." In Learning for Dynamics and Control, pp. 256-266. PMLR, 2020.

---

> > > ### Author Response · Authors · 2021-12-06
> > > **Official Response**
> > >
> > > We thank the reviewer for his careful considerations and remarks.
> > >
> > > Regarding Proposition 2, we will indeed remove the words "to a GMFE", which are no longer of importance in view of Theorems 4 and 5, as we have shown there that we obtain the same $(\epsilon, p)$-Nash property as a GMFE even without converging to an exact GMFE. Regarding the other comments, 1. the notation in Theorem 4 is consistent with the rest of the work, 2. the $\hat \Phi$ in Theorem 5 has been defined before in Section 4, paragraph "Equivalence classes method" as "the optimal policy of $\alpha_i$ for fixed $\boldsymbol \mu$", and 3. in our opinion the algorithms are clearly described, see in particular the text as well as Algorithms 1, 2, and 3 in the Appendix in addition to the main text (Section 4).
> > >
> > > Regarding MAPPO and MARL, although the referenced paper in particular investigates performance of PPO via independent learning in the cooperative case, note that PPO is similarly applied to complex competitive MARL problems, e.g. in [A], and the value of centralised critics is sometimes even disputed, e.g. in [B]. Additional considerations are that the complexity of centralized critics rises sharply with the number of agents, as well as that a theoretical foundation of such centralized critic methods is typically missing.
> > >
> > > More importantly, as mentioned in our previous comment, we still believe that our Theorems 4 and 5 are sufficiently convincing, as the limiting optimality of our approach has been rigorously shown and thus does not require an empirical comparison to approximate MARL methods. Instead, we focus on the limiting system, as it is typically done to the best of our knowledge in all of the existing literature on learning MFGs, e.g. [C, D, E] or any other reference in our manuscript. We do not believe that this diminishes the theoretical and algorithmic contributions of our work.
> > >
> > > Note that the results we obtain are exact and rigorous, and we believe that our work has only improved over the first, pre-rebuttal version.
> > >
> > > [A] Berner, Christopher, et al. "Dota 2 with large scale deep reinforcement learning." arXiv preprint arXiv:1912.06680 (2019).
> > >
> > > [B] de Witt, Christian Schroeder, et al. "Is Independent Learning All You Need in the StarCraft Multi-Agent Challenge?." arXiv preprint arXiv:2011.09533 (2020).
> > >
> > > [C] Subramanian, Jayakumar, and Aditya Mahajan. "Reinforcement learning in stationary mean-field games." Proceedings of the 18th International Conference on Autonomous Agents and MultiAgent Systems. 2019.
> > >
> > > [D] Elie, R., Perolat, J., Laurière, M., Geist, M., & Pietquin, O. (2020, April). On the convergence of model free learning in mean field games. In Proceedings of the AAAI Conference on Artificial Intelligence (Vol. 34, No. 05, pp. 7143-7150).
> > >
> > > [E] Guo, Xin, et al. "Learning Mean-Field Games." Advances in Neural Information Processing Systems 32 (2019): 4966-4976.

---

> ### Author Response · Authors · 2021-11-17
> **Official Response to Review of Paper1361 by Reviewer GRFn (2/3)**
>
> ...
>
> > 3. The numerical experiments lack some important comparisons. Firstly, since GMFG is proposed to solve N-player games approximately, existing algorithms for solving N-player games (especially those proposed for multi-agent reinforcement learning) are all fair comparisons. Hence to demonstrate the effectiveness of the proposed two learning schemes, comparisons with algorithms such as those mentioned in (Yang et al., 2018) and [A] are needed. Also, performance in the N-player setting is only demonstrated with the indirect metric (26), which is not sufficiently convincing and informative.
>
> We agree partially, though we would like to point out that our work focuses on using RL to solve the limiting system, and does not yet constitute a MARL method to be applied to arbitrary finite-agent systems. The important difference here is that the limiting mean field problem only models a representative agent (with graphon index as state) together with a mean field term, i.e. we apply single-agent reinforcement learning to solve the limiting system, more similar to works such as [D, E, A] rather than (Yang et al., 2018). For MARL, the additional step of graphon estimation is an important step, as discussed in the conclusion and Section 4.
>
> We have further added a formal proof of optimality for our equivalence classes method (Theorem 5). Together with the added ablation over the number $M$ of equivalence classes, which shows stability of our proposed solution over $M$, we believe our analysis to suffice, as there would be little point in comparing approximate MARL methods with an optimal solution. Nonetheless, we have added an exemplary experiment using MARL methods at the end of the appendix, where due to the non-stationarity of the other agents, a naive application of state-of-the-art MARL techniques fails to converge, while the mean field MARL approach in (Yang et al., 2018) remains incomparable due to necessity of observing the average previous numerical actions of other agents, whereas our setting remains fully decentralized (local state only).
>
> Regarding the interpretation of the N-agent comparison experiment, the performance in the N-agent system is problematic to assess due to the multitude of reasons discussed in Section 5, and has been rigorously shown in Theorem 3 already, which obviates the strict need of empirical verification. Instead, the experiment verifies the accuracy of the mean field system as an approximation to the finite-agent system. To better reflect this, we have reformulated the paragraph accordingly.
>
> ## Regarding the minor comments and questions
>
> > 1. The authors should compare with [B].
>
> We deeply appreciate and thank the reviewer for pointing out relevant and important related work, and have added a corresponding comparison in the related work section. In contrast to our work, the authors in [B] are missing a theoretical analysis of approximation properties such as the approximation of finite graph systems and reduction to classical MFGs, while on the empirical side [B] constructs recursive equations for an infinite-dimensional (and thus difficult to compute in general) value function defined over all possible mean field ensembles, which are therefore not immediately applicable to arbitrary problems in a black-box, learning manner.
>
> > 2. On page 2, should “Graph mean field systems” be “Graphon mean field systems”?
>
> With "graph mean field systems" we meant mean field systems on general graphs, as we discuss not only graphon-based approaches, but also some studies of mean field systems on sparser graphs. To make it more clear, we have renamed the section to "mean field systems on graphs".
>
> > 3. Please discuss more about the relationship between the GMFG model and the one adopted in (Yang et al., 2018) and [C], which are also related to graph based games.
>
> We would like to point out that the theory of MFMARL (Yang et al., 2018) holds only under restrictive assumptions, and MFMARL averages only over neighbor's actions instead of depending on the full distribution, as also pointed out e.g. by [D]. Additionally, conceptually MFMARL is a MARL algorithm assuming knowledge of neighboring agent actions, whereas our work applies RL to limiting graphon mean field games and remains fully decentralized, in that an agent only reacts to its own state. In [C], the graph structure is over states instead of agents, and the inverse reinforcement learning setting is considered. We have added the corresponding discussion to the introductory section.
>
> ...

---

> ### Author Response · Authors · 2021-11-17
> **Official Response to Review of Paper1361 by Reviewer GRFn (1/3)**
>
> We thank the reviewer for their detailed reading as well as highly constructive comments. We believe that the points raised by the reviewer have been addressed, in turn significantly strengthening our paper as well as our algorithmic contribution. Please note that the length of the paper has increased by 6 pages in order to address all feedback.
>
> ## Regarding the major concerns
>
> > 1. Section 2 is a bit too heavy as an introductory section. It might be better to start with Section 2.1, and describe the idea of local interaction and local averaging for a group of networked agents in a more intuitive manner. Some visualization of the N-agent scenario will also be helpful (see e.g., (Yang et al., 2018)).
>
> We agree and thank the reviewer for the very constructive suggestion. We have added a simple visualization of the graphical game dynamics (see Figure 1) as well as further introductory words at the beginning of section 2 and the introduction of the finite agent game.
>
> > 2. The algorithm description and motivation of the first learning method (which discretizes the graphon index) are not sufficiently clear. It would be helpful to compare this discretization approach with multi-population mean-field games in more details and discuss about what are the benefits of considering GMFG, and to better explain why for each equivalence class the optimal control problems can be solved independently. The verbal description is a bit too informal, while the Appendix A.3 is too focused on the experiment details while not providing sufficient explanations on how these heuristic algorithms are derived. The justification of the algorithm design logic is especially important since no formal convergence theory for the proposed learning algorithms is established.
>
> We agree on the point of algorithm description and motivation, and have added a more detailed, formal justification at the end of Section 4 with proof (Theorem 5): We note that under our assumptions of Lipschitz continuous $W$, the difference between neighborhoods (11) of all agents in an equivalence class tends to zero. Thus, their dynamics become increasingly similar, which implies that the optimal policy of the representative $\alpha_i$ becomes near-optimal for all $\alpha$ in the same equivalence class. Similarly, the mean fields become arbitrarily close to equal. Thus, a solution of the equivalence class approach will fulfill the Nash property arbitrarily well.
>
> The benefit of GMFGs over multi-class MFGs is their rigorous connection to the finite graph games, as well as their ability to handle "infinitely many" classes, whereas classical multi-class MFGs and their theory so far remain restricted to a finite number of classes and fail to account for graphical structure. The corresponding discussion was added to Section 4.
>
> The optimal control problem of each equivalence class for a fixed mean field is independent of each other only during one iteration of the algorithm, since in the limiting formulation each representative agent only interacts with the mean field. This does not mean that the final solution of one equivalence class does not consider the behavior of other equivalence classes, since at convergence the fixed point property (optimality) must hold for all agents at once.
>
> A rigorous justification for the design of our algorithms has been added in Section 4, together with a formal convergence theory for the equivalence class approach via Theorem 4. An analysis of the more elegant direct reinforcement learning approach remains outside of the scope of our work, as significant further analysis would be required.
>
> ...

---

### Author Response · Authors · 2021-11-17
**Manuscript Update**

We deeply thank all reviewers for their detailed reading as well as highly useful feedback. We believe that the points raised by all reviewers have been addressed, in turn significantly strengthening our paper as well as our algorithmic contribution. Please note that the length of the paper has increased by 6 pages in order to address all feedback.

In response to the feedback received by all reviewers, we have performed the following list of major changes:

- We have added a rigorous proof of optimality for our equivalence classes method for sufficiently fine discretization, implying that a solution of the equivalence class approach will fulfill the Nash property arbitrarily well.
- We have added a formal convergence theory for our discretization approach.
- We have added an improved explanation and visualization of the setting in Section 2.
- We have added additional experiments, ablating over the number $M$ of discretization intervals and showing the stability of our algorithm over $M$.
- We have added discussions based on feedback of all reviewers, most notably 1. further discussion of related works, 2. the benefit of GMFGs over multi-class MFGs, 3. the difference to MARL methods, 4. additional motivation for the reinforcement learning method, and a great number of smaller clarifications.
- We have added further related work together with discussions.
- We have improved equation formatting and headings to save space. For space reasons, some figures, captions and paragraphs have been slightly resized or moved to the appendix to incorporate the feedback from reviewers.
- We have updated the code (supplementary material) accordingly.

---

### Decision · Program_Chairs · 2022-01-20

**Decision:**

Accept (Poster)

**Comment:**

This paper studies graphon mean-field games, whereby a continuum of agents are connected by a graphon. They study a discrete time version and show existence of a Nash equilibrium (under Lipschitz conditions). Moreover they prove that it corresponds to an approximate Nash equilibrium for the game with a finite number of players, thereby validating graphon mean-field games as a natural abstraction when the number of players is sufficiently large. Finally they give algorithms based on fixed point iterations (one based on discretizing the graphon index, the other based on reformulating it as a classical mean-field game) for computing such an equilibrium. They give numerical experiments to validate their approach. The reviewers pointed out various writing issues or other results that would help complete the picture. Many of these were addressed and/or clarified by the authors in their revision. Overall the paper provides an appealing and relatively complete characterization of equilibria in graphon mean-field games.